# Nucleotide-dependent switching and RIPb effector recognition of the barley susceptibility factor RACB
Mariam Mohamadi[1], Mariem Bradai [2], Robert Janowski[3], Umut Günsel [1,3], Marie Tran[1], Sinika M. Kahl[1], Christopher McCollum[2], Dierk Niessing[3,4], Ralph Hückelhoven [2] & Franz Hagn [1,3]✉

ROP proteins are plant-specific members of the Rho family of small GTPases that orchestrate fundamental signaling pathways controlling cell polarity, directional growth, and immune responses. Although their biological importance is well established, the structural basis underlying their activation and interactions with downstream effectors has remained insufficiently understood. Here, we present an atomic-resolution structural analysis of RACB, a ROP GTPase from barley (*Hordeum vulgare*) that functions as a key susceptibility factor during fungal infection. Using an integrative approach combining X-ray crystallography, nuclear magnetic resonance spectroscopy, and hydrogen-deuterium exchange mass spectrometry, we capture high-resolution structural and dynamical snapshots of RACB in both its inactive and active conformations. This setup reveals the conformational flexibility and switching mechanism that are central to RACB function. Moreover, the structure of the complex between active RACB and its effector protein RIPb uncovers the fully activated state of RACB and identifies a conserved interaction motif within RIPb that mediates complex formation, providing mechanistic insights into how RIPb can link membrane-associated RACB to the microtubule cytoskeleton to facilitate membrane remodeling processes. These findings establish a detailed structural framework for plant Rho-type GTPase signaling and offer a molecular explanation for how pathogens exploit ROP-mediated pathways to promote infection in plants.

Small monomeric G proteins, such as Rho GTPases, play a crucial role in regulating various cellular processes across all eukaryotic kingdoms. GTPases can switch between an inactive (GDP-bound) and active (GTP-bound) state, with conformational changes upon GTP binding allowing them to engage with downstream effector proteins and trigger signaling pathways. The cycle between these states is tightly controlled by guanine nucleotide exchange factors (GEFs), which facilitate the exchange of GDP for GTP, and GTPase-activating proteins (GAPs), which enhance the GTPase activity, resulting in GTP hydrolysis and inactivation[1,2] (Fig. 1a). Small GTPases have a conserved G-domain, which consists of the five G-motifs at the N terminus: G1 (GxxxxGK[S/T]) or P-loop, G2 (Thr), G3 (DxxGQ), G4 (NKxD), and G5 (TSAK)[3] (Supplementary Fig. 1). In animals and fungi, the well-characterized Rho GTPase family members regulate diverse cellular processes through interactions with specific downstream effectors[4]. Plants only have one

distinct and essential subfamily of Rho GTPases called RAC (rat sarcoma-related C3 botulinum toxin substrate) or ROP (Rho of plants)[5,6].

RAC/ROPs have evolved to regulate a unique set of plant-specific processes, such as cell polarity and tip growth, which are fundamental to the development of pollen tubes and root hairs[7,8]. These proteins guide cell shape and morphogenesis by directly interacting with actin-binding proteins[9,10]. ROPs mediate responses to both biotic and abiotic stresses[11,12] and are essential players in plant immunity and defense response[13,14]. This functional versatility is achieved through the activation of a diverse array of unique downstream effectors to induce specific cellular responses[15]. However, compared to the human counterparts, relatively few studies have focused on ROP GTPases in plants. Although 11 RAC/ROPs have been identified in *Arabidopsis thaliana*[16], only a limited number are characterized structurally[17–20]. RAC/ROPs are classified into two sub-types according to

[1]Bavarian NMR Center (BNMRZ) and Structural Membrane Biochemistry, Dept. of Bioscience, TUM School of Natural Sciences, Technical University of Munich, Garching, Germany. [2]Chair of Phytopathology, TUM School of Life Sciences, Technical University of Munich, Freising, Germany. [3]Molecular Targets and Therapeutics Center (MTTC), Institute of Structural Biology, Helmholtz Munich, Neuherberg, Germany. [4]Institute of Pharmaceutical Biotechnology, Ulm University, Ulm, Germany. ✉e-mail: franz.hagn@tum.de

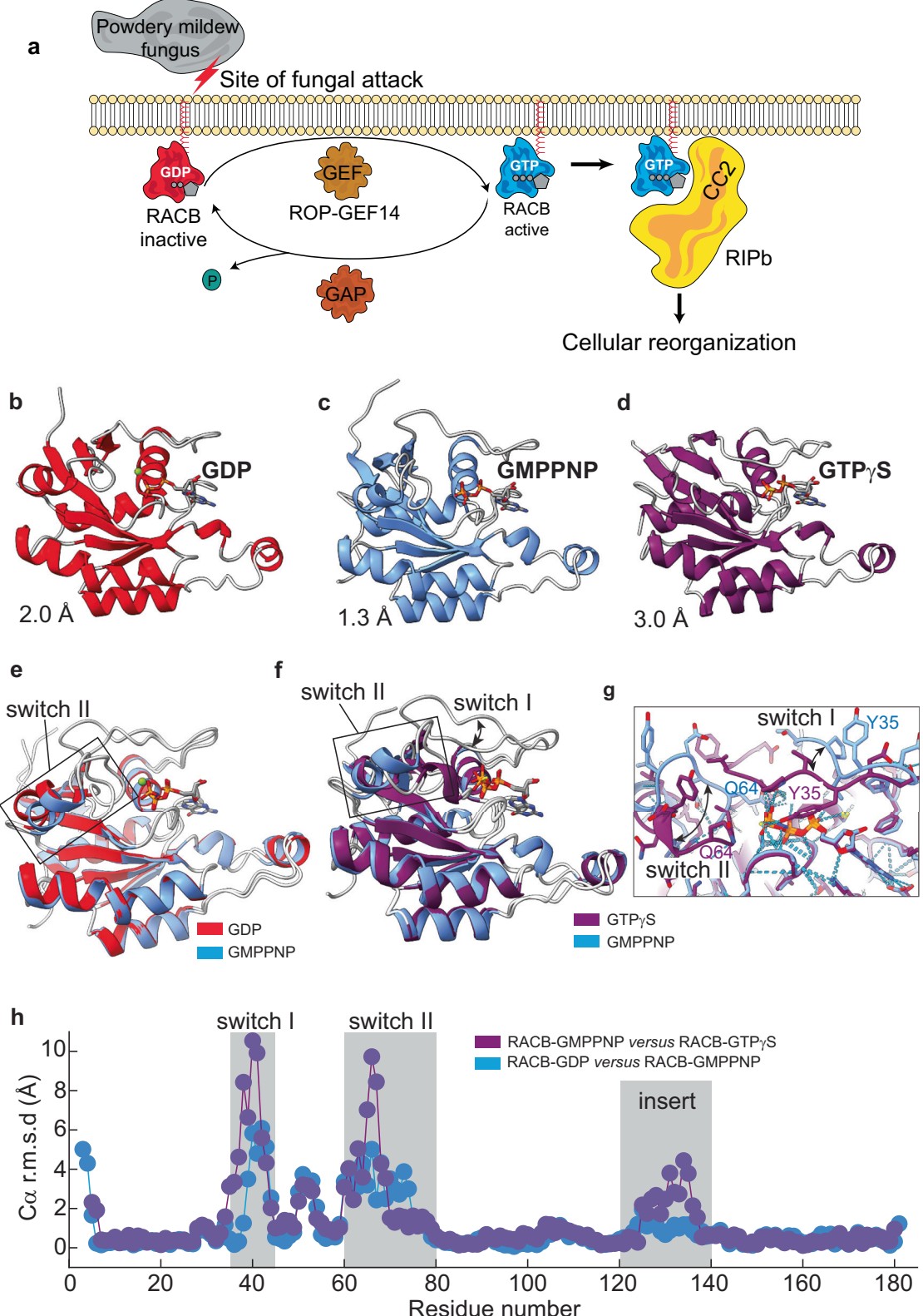

the presence of a C-terminal CXXL or CaaX-box (type I, X: aliphatic amino acid) or internal cysteine motifs, such as GC-CG (type II)[6,21]. In type I ROPs, such as RACB, the C-terminal cysteine is irreversibly farnesylated, followed by proteolytic removal of the aaX residues and carboxymethylation[22], whereas type II ROPs undergo reversible S-acylation (palmitoylation) on internal cysteines[14,23]. These lipid modifications are essential for anchoring ROPs to the plasma membrane[21,24]. Additionally, the C-terminus contains a

hypervariable region (HVR) with a polybasic stretch of lysine residues, which supports binding to negatively charged membrane surface patches[25], where interactions with effector proteins take place. Like their mammalian equivalents, ROPs feature an insert region, a unique characteristic feature of Rho proteins that is important for effector protein binding[26,27].

RACB, a type I ROP GTPase in barley (*Hordeum vulgare*)[28], has been originally identified as a RAC1 homolog with 98% identity to orthologous

**Fig. 1 | Structures of RACB in different nucleotide-bound states demonstrate its switching functionality required for membrane remodeling in plants upon fungal attack. a** Schematic representation of the RACB activation cycle and its role during powdery mildew infection. At the site of fungal attack, the small GTPase RACB transitions between an inactive GDP-bound form and an active GTP-bound form, potentially enhanced by ROP-GEF14[33]. Active RACB interacts with downstream effectors such as RIPb, which mediates cellular reorganization that supports fungal penetration. GEF: guanine nucleotide exchange factor; GAP: GTPase activating protein. **b** Crystal structures of RACB-GDP (red) at 2.0 Å resolution (PDB ID: 9T3C), **c** of RACB- GMPPNP (blue) at 1.3 Å resolution (PDB ID: 9T3D) and

**d** RACB-GTPγS at 3.0 Å resolution (PDB ID: 28NN). **e** Structural overlay of RACB-GDP (red) and RACB-GMPPNP (blue), highlighting conformational rearrangements in switch II. **f** Structural overlay of RACB-GMPPNP (blue) and RACB-GTPγS (purple), showing conformational changes in both switch regions. **g** Close-up view of the nucleotide binding site in the structural overlay shown in (**f**) where the RACB-GTPγS adopts a more compact structure with the two switch regions in close proximity. **h** Cα displacement plot illustrating backbone deviations between RACB-GDP and RACB-GMPPNP (blue) and between RACB-GMPPNP and RACB-GTPγS (purple). The most significant changes occur in switch I, II, and the insert region (gray boxes).

proteins in rice and maize. It is located at the plasma membrane by a C-terminal farnesyl moiety and a polybasic stretch[29]. In barley, RACB has emerged as a key player in cell polarity, and importantly, in the plant's susceptibility to the fungal biotrophic pathogen *Blumeria hordei (Bh)*. Studies showed that constitutively activated RACB (CA RACB, G15V) significantly enhances the successful penetration of *Bh*, whereas silencing of RACB reduces the infection rate[30–32]. During pathogen attack, host cells undergo major structural changes that help *Bh* penetration. In the early infection stages, the fungal parasite might activate RACB and its downstream effector proteins via a plant-specific ROP-GEF14[33]. This cascade alters the organization of the host cell's cytoskeleton to control vesicle trafficking and fusion[34–36] and penetrate the host cell wall, which creates a defined apoplastic compartment around the haustorium[37,38] (Fig. 1a).

An important class of effector proteins involved in that process is the interactor of constitutive active ROP (ICR) or ROP Interactive Partners (RIPs). ICR/RIPs are a distinct but poorly characterized class of plant-specific proteins that contain coiled-coil domains, functioning as scaffolding proteins in ROP signaling pathways[39,40]. It has been shown that RIPb directly interacts with active RACB and enhances fungal penetration into epidermal cells[41]. Upon RACB activation, RIPb is recruited to the plasma membrane, where it co-localizes with the active GTPase. Furthermore, like other RIPs[42], RIPb is observed to associate with microtubules and is specifically targeted to the plasma membrane at the host-pathogen interface at the site of fungal attack[41]. Phylogenetic analysis of RIP proteins in barley revealed a characteristic peptide motif (termed QWRKAA) within the predicted C-terminal coiled-coil domain of RIPs, which is assumed to be the primary interaction site for ROPs[39]. In line with this finding, yeast two-hybrid experiments showed that the C-terminal coiled-coil domain of RIPb (RIPb-CC2) is sufficient to bind to active RACB[41]. Dysregulated ROP activity has been associated with various plant diseases, highlighting the importance of understanding their signaling pathways and regulatory mechanisms[37]. However, a detailed understanding of the switching mechanism and associated dynamics underlying barley RACB function and its interaction with downstream effector proteins remains elusive so far.

Here, we present high-resolution structures of the barley ROP GTPase RACB in its active and inactive states, as well as in the complex with its effector protein RIPb. These structures, together with NMR and biophysical analyses, reveal how nucleotide-dependent switching impacts the conformational landscape of RACB to enable allosteric communication between the nucleotide binding site and the effector interface. Binding assays, yeast two-hybrid, and *in planta* fluorescence complementation studies corroborate the conserved QWRKAA motif as the critical element of ICR/RIP-ROP recognition. Collectively, our findings provide atomistic insights into RACB's conformational switching dynamics and effector recognition. This structural framework advances our understanding of ROP-mediated signaling in plants and provides valuable insights into the role of RIPb as a scaffold protein that connects membrane-bound RACB with the microtubule cytoskeleton for membrane remodeling.

## Results
### Structural characterization of the nucleotide-dependent switching functionality of the barley ROP GTPase RACB
For this study, we obtained well-folded and highly pure proteins by recombinant expression in *E. coli* with a cleavable N-terminal GB1 fusion

tag (Supplementary Fig. 2a–c). We first set out to analyze the structural features of RACB bound to three different nucleotides (GDP, GMPPNP, GTPγS) by circular dichroism (CD) spectroscopy. Far-UV CD spectra of all three nucleotide-bound RACB samples indicated a characteristic small GTPase α/β-secondary structure topology with no striking differences in the different nucleotide-bound states (Supplementary Fig. 3a) with very similar thermal stabilities of RACB bound to GDP and GMPPNP ($T_m$ = 44 °C) but a markedly increased thermal stability in the GTPγS-bound state ($T_m$ = 52 °C) (Supplementary Fig. 3b), similar to what has been previously observed for a G-protein α-subunit[43].

RACB bound to the three different nucleotides showed well-dispersed 2D-[$^{15}$N, $^{1}$H]-HSQC NMR spectra (Supplementary Fig. 3c) indicating a compactly folded protein in each case. However, noticeable chemical shift changes distinguish the inactive GDP-bound form from the two active GTP analog-bound states. The most significant chemical shift perturbations and line broadening effects were observed with GTPγS, in line with its distinct thermal stability compared to the other nucleotides. To analyze the structural basis of its nucleotide-dependent switching behavior, we determined crystal structures of wild-type RACB (1–197) in complex with GDP, GMPPNP and GTPγS at resolutions of 2.2 Å, 1.9 Å and 3.0 Å, respectively (Table 1, Fig. 1b–d).

The structures reveal that RACB adopts a typical small GTPase fold, comprising a core of six β-strands surrounded by five α-helices. RACB contains characteristic conserved structural motifs of RAC/ROP GTPases[44], such as the G1/P-loop (G$_{13}$DGAVGKT$_{20}$), involved in phosphate binding and GTP hydrolysis; switch I (Y$_{35}$VPTVFDNFSAN$_{45}$), containing Thr38, which coordinates Mg$^{2+}$ and plays a role in nucleotide sensing and effector binding; and switch II (L$_{58}$WDTAGQEDYNRLRPLSYRGADV$_{80}$), essential for GTP hydrolysis. Additional conserved elements include the G4 (L$_{115}$VGTKLDLR$_{123}$) and G5 motifs (Y$_{155}$IECSSKT$_{162}$), both of which are critical for guanine nucleotide recognition. A distinctive feature of RACB is the Rho insert region (residues 124-140), a unique effector loop that is absent in Ras proteins but conserved among Rho-family GTPases[44]. The membrane attachment site, comprising the hypervariable polybasic region and the C-terminal CSIL motif (176-197), is not visible in either crystal structure due to the high flexibility of this unfolded C-terminal tail. With the GDP- and GTP-analogue-bound structures of RACB determined here, an accurate assessment of the nucleotide-dependent conformational changes during activation is possible.

Pronounced conformational differences between the GDP- and GMPPNP-bound structures can be observed mainly in switch II (Fig. 1e, h). Furthermore, the N-terminal β-sheet structure is well-ordered with a stable β-strand arrangement if bound to GDP but shows a more disordered state in the GMPPNP-bound form. A part of the switch I region (residues 44-47) forms an extended β-strand in the GDP-bound state that helps maintain the compact structure of the nucleotide-binding pocket. In the GMPPNP-bound state, the loop region in switch I is longer, suggesting a more open conformation. Interestingly, the α-helical secondary structure of switch II (residues 66-69) is better defined when bound to GMPPNP, indicating stabilization of this region by the additional phosphate moiety. The insert region, which is partly α-helical, does not show conformational changes between the two structures. For RACB-GMPPNP, no electron density for the coordinated Mg$^{2+}$ ion was visible since the Mg$^{2+}$ binding site is occupied by the side chain of Lys182 from a symmetry-related molecule in the

**Table 1 | Data collection and refinement statistics for RACB complexes**

| | RACB +GDP PDB ID 9T3C | RACB +GPPNHP PDB ID 9T3D | RACB +GTPγS PDB ID 28NN | RACB + GTPγS + RIPb-CC2 Parallel PDB ID 9T3F | RACB + GTPγS + RIPb-CC2 Antiparallel PDB ID 9T3E |
|---|---|---|---|---|---|
| **Data collection** | | | | | |
| Space group | $P\,3_1\,2$ | $C\,2$ | $C\,2\,2\,2_1$ | $P\,6_1\,2\,2$ | $C\,2\,2\,2_1$ |
| Cell dimensions | | | | | |
| $a, b, c$ (Å) | 98.32, 98.32, 107.05 | 85.71, 66.70, 37.10 | 110.24, 132.51, 196.06 | 64.00, 64.00, 243.32 | 76.51, 139.45, 60.11 |
| α, β, γ (°) | 90.00, 90.00, 120.00 | 90.00, 93.59, 90.00 | 90.00, 90.00, 90.00 | 90.00, 90.00, 120.00 | 90.00, 90.00, 90.00 |
| Resolution (Å) | 50–2.03 (2.15–2.03) | 50–1.33 (1.36–1.33) | 50–3.00 (3.08–3.00) | 50–2.07 (2.12–2.07) | 50–2.30 (2.36–2.30) |
| $R_{sym}$ | 5.8 (83.4) | 4.7 (62.3) | 16.0 (233.7) | 6.4 (180.1) | 6.9 (175.1) |
| $I\,/\,\sigma I$ | 9.7 (1.2) | 21.5 (3.2) | 12.1 (1.0) | 24.5 (1.1) | 19.5 (1.0) |
| Completeness (%) | 99.8 (99.6) | 99.0 (97.5) | 99.9 (99.8) | 99.9 (99.9) | 99.9 (100) |
| Redundancy | 8.1 (7.7) | 6.8 (6.0) | 13.7 (12.3) | 19.1 (20.2) | 13.4 (13.2) |
| **Refinement** | | | | | |
| Resolution (Å) | 2.03 | 1.33 | 3.00 | 2.07 | 2.30 |
| No. reflections | 39,262 | 47,328 | 29,045 | 18,023 | 14,643 |
| $R_{work}\,/\,R_{free}$ | 17.26 / 19.66 | 11.64 / 14.00 | 21.48 / 27.34 | 20.55 / 24.69 | 25.85 / 31.75 |
| No. atoms | | | | | |
| Protein | 2,873 | 1,701 | 8,258 | 1,555 | 1,841 |
| Ligand/ion | 123 | 54 | 199 | 33 | 44 |
| Water | 169 | 214 | 13 | 51 | 14 |
| $B$-factors | | | | | |
| Protein | 52.1 | 19.8 | 113.9 | 76.9 | 92.5 |
| Ligand/ion | 60.4 | 23.1 | 108.1 | 47.2 | 97.2 |
| Water | 56.6 | 37.3 | 94.9 | 61.6 | 86.5 |
| R.m.s. deviations | | | | | |
| Bond lengths (Å) | 0.019 | 0.011 | 0.009 | 0.008 | 0.008 |
| Bond angles (°) | 1.82 | 1.85 | 1.12 | 1.67 | 1.09 |

*For each structure, diffraction data for a single crystal were used. *Values in parentheses are for highest-resolution shell.

asymmetric unit. In line with the marked differences in thermal stability, the structure of RACB-GTPγS (Fig. 1d) shows pronounced changes when compared to RACB-GMPPNP (Fig. 1f–h). Both switch regions and the insert region are altered with a more closed structure occupied with GTPγS where e.g., switch I residue Y35 moving towards the γ-phosphate of the nucleotide and switch II adopting an α-helical conformation with Q64 moving away from the nucleotide and closer to the opposing switch I (Fig. 1f, g).

To understand how RACB's conformational switching signature aligns with other plant Rho GTPases, we compared our structures with the few available ROP structures. Due to a lack of structures of the same ROP with different nucleotides, we used the *Arabidopsis thaliana* GDP-bound ROP9 (PDB ID: 2J0V[18]) and *Oryza sativa* (rice) GMPPNP-bound RAC1 (PDB ID: 4U5X[17]) (Supplementary Fig. 4a) for a structural comparison. The analysis revealed larger structural differences as described in the RACB case (high root mean square deviations, r.m.s.d. in Supplementary Fig. 4b). Mainly switch I and, to an even larger extent, switch II show pronounced conformational changes. In switch II, some residues in ROP9-GDP are not resolved in the crystal structure, presumably due to high flexibility. Furthermore, switch II in ROP9-GDP does not adopt a regular helical secondary structure, unlike the well-defined helical switch II region in RAC1-GMPPNP. The insert region also shows structural differences. However, such a difference outside the switch regions can be attributed to the use of different small GTPases in this comparison. In summary, RACB undergoes stepwise conformational changes from the GDP to the GMPPNP- and

GTPγS bound states, where the switch regions as well as neighboring β-strand geometries are affected.

## RACB adopts a dynamic conformational state upon activation
Given that our RACB X-ray structures show nucleotide-dependent conformational changes, we sought to determine whether these structural changes and associated alterations in dynamics can also be detected in solution. Therefore, we first set out to obtain sequence-specific NMR backbone resonance assignments of wild-type RACB (1–197) bound to GDP or the non-hydrolyzable GTP analog GMPPNP. We used GMPPNP instead of GTPγS, since the number of visible peaks in the 2D-NMR spectrum was markedly higher (Supplementary Fig. 3c). Using a set of triple-resonance NMR experiments[45], we successfully assigned 82% of the non-proline residues in RACB-GDP and 80% in RACB-GMPPNP (Supplementary Fig. 5a, c, Supplementary Data 1). The few unassigned residues are in switch I (34-44) and switch II (68-74), as well as in the flexible C-terminal tail. GMPPNP-bound RACB shows a higher degree of line broadening and a lower number of visible NMR resonances, presumably caused by dynamics in the μs to ms time scale. The secondary structure content of RACB in both samples derived from NMR secondary chemical shift values (Supplementary Fig. 5b, d) are consistent with the obtained X-ray structures (gray shaded areas in Supplementary Fig. 5b, d).

Due to many missing backbone amide signals, RACB-GTPγS was not used for NMR resonance assignment experiments. However, the visible resonances in the 2D-NMR spectrum (Supplementary Fig. 3c) of RACB-

**Fig. 2 | Hydrogen-deuterium exchange (HDX) mass spectrometry and NMR chemical shift perturbation (CSP) analysis of active and inactive RACB. a** Deuterium uptake of RACB-GDP (red) and RACB-GTPγS (blue) states plotted against the residue number. Shaded regions indicate the conserved GTPase motifs P-loop, switch I, and switch II. Peptide coverage is depicted in Supplementary Fig. 6. **b** HDX difference plot of GTPγS-*versus* GDP-bound RACB mapped onto the crystal structure of RACB-GDP and color-coded by deuterium uptake as indicated in the legend. **c, d** Chemical shift perturbations of RACB-GDP *versus* RACB-GMPPNP plotted against the residue number and color-coded on the RACB-GDP crystal structure. Residues colored in red (see legend) show the most significant values (dotted line: mean plus two standard deviations). Negative values (orange) indicate signal loss in GMPPNP-bound RACB, while purple bars show additional signal loss with GTPγS (see Supplementary Fig. 7).

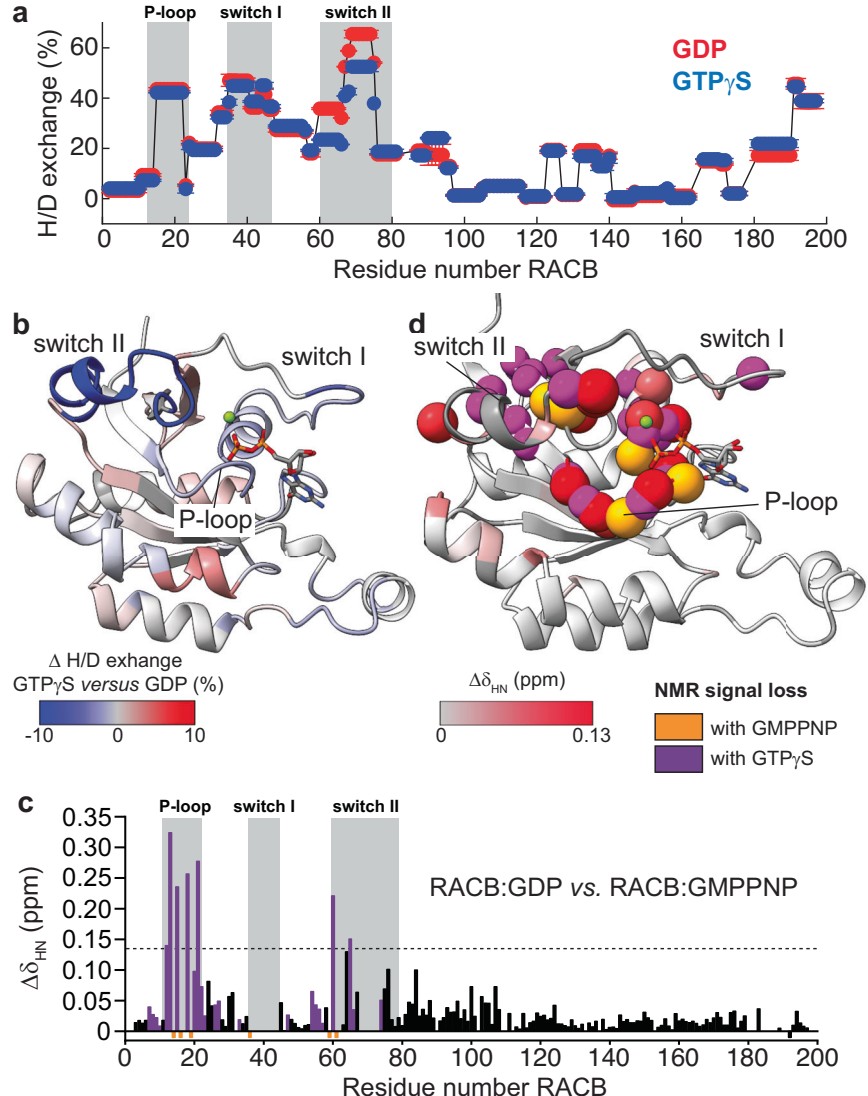

GTPγS overlay very well with most resonances of the GMPPNP-bound form, suggesting a similar structure but enhanced dynamics in μs-ms time scale. To gather further structural information about the GTPγS-bound state, including the NMR-invisible regions, we used hydrogen-deuterium exchange mass spectrometry (HDX-MS) with RACB bound to GDP or GTPγS. HDX-MS detects the incorporation of deuterium in backbone amides, providing a sensitive readout of local hydrogen-bond stability and solvent accessibility[46]. Regions with faster deuterium uptake correspond to more flexible or solvent-exposed regions of the protein[47,48]. A comparison of the HDX-MS profiles of GDP- (red symbols) and GTPγS-bound (blue symbols) RACB (Fig. 2a) revealed a higher level of H/D exchange for RACB-GDP as compared to RACB-GTPγS in switch II (58-68) with only slight differences in other regions (Fig. 2b).

To gain further insights on the conformational changes in RACB at the per-residue level, we analyzed NMR backbone amide chemical shift perturbations (CSPs) between GDP- and GMPPNP-bound RACB (Fig. 2c, d). The largest CSPs mapped to the nucleotide-binding site, involving the P-loop and switch II, which confirmed the effect of GMPPNP on increasing the α-helical content in switch II as observed in our crystal structures (Fig. 1b, c, e). All missing resonances in the GMPPNP-bound state map to the P-loop and the two switch regions (orange bars in Fig. 2c and orange spheres in Fig. 2d), suggesting that the GTP-analog enhances conformational exchange dynamics as compared to GDP. Since NMR line broadening

caused by μs-ms dynamics appears to be a hallmark for RACB activation, we further analyzed the 2D-NMR spectrum of RACB-GTPγS in more detail (Supplementary Fig. 7a). With the NMR chemical shift assignments obtained in the complex with GMPPNP, we could identify the amino acid positions of missing resonances in RACB-GTPγS and found 27 additionally missing peaks corresponding to residues located in the P-loop, switch I and II (magenta bars in Fig. 2c and magenta spheres in Fig. 2d), consistent with the effects seen by HDX-MS (Fig. 2a, b).

### NMR-detected dynamics of RACB at multiple time scales

Since dynamical features appear to be a critical factor in RACB activation, we set out to quantify the dynamics of RACB at multiple time scales. RACB is a monomer as seen in SEC at an estimated peak protein concentration of 35 μM (Supplementary Fig. 2). First, we probed fast internal motions in the nanosecond-to-picosecond time scale by recording steady-state {¹H}-¹⁵N heteronuclear Overhauser effect (hetNOE) experiments at a magnetic field strength corresponding to 950 MHz ¹H frequency, for RACB-GDP and RACB-GMPPNP (Supplementary Fig. 8a, b). Values around 0.8 indicate a rigid backbone amide, while lower values suggest the presence of internal motions on the ns-ps timescale. While the hetNOE values follow a similar pattern in both samples (Supplementary Fig. 8c, d), we nonetheless could detect differences at the N-terminus, where RACB-GMPPNP has lower hetNOE values (faster ns-ps motions) and in switch II, where the GMPPNP-

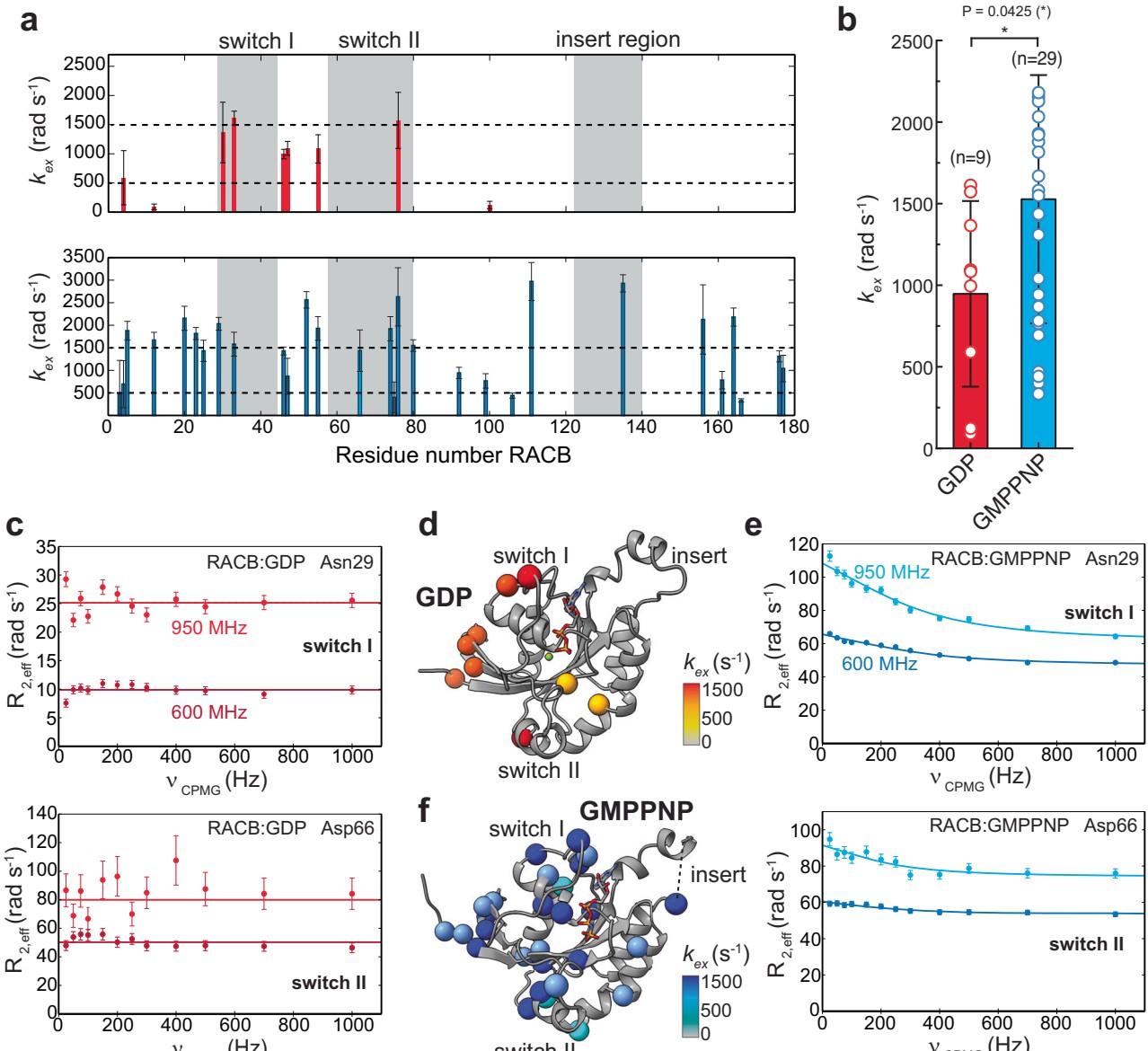

**Fig. 3 | Conformational exchange dynamics of RACB probed by $^{15}$N CPMG relaxation dispersion experiments. a** Relaxation dispersion rates plotted against the residue number for RACB-GDP (red) and RACB-GMPPNP (blue). Error bars represent the fitted error. **b** Bar plot of the individual exchange rates with the bar height representing the mean value and S.D. the error bar. * $p < 0.05$ (**c**) and (**e**) representative fits of dispersion curves at 600 and 950 MHz for RACB-GDP (red) and RACB-GMPPNP (blue). Error bars are obtained by error extrapolation based on three duplicate experiments at 75, 200, and 700 Hz $^{15}$N CPMG frequencies, (**d**) and (**f**) structural mapping of residues in GDP or GMPPNP-bound RACB showing μs-ms motions.

bound form exhibits a lower degree of fast motions as indicated by the higher hetNOE values. The effect of GMPPNP on the intrinsic dynamics in switch II is consistent with the higher α-helical content in the X-ray structure (Fig. 1b, c, e).

To connect these experimental dynamical parameters with the structural features as seen in the X-ray structures, we conducted three independent molecular dynamics simulations (10 μs) for each state, and quantified the observed motions by the root mean square fluctuation (RMSF) of Cα atoms in RACB-GDP and RACB-GMPPNP (Supplementary Fig. 8a, b, lower panels). These RMSF patterns agree very well with the experimental hetNOE profiles with switch I and II, and the insert region showing the highest degrees of motion, where GMPPNP tunes down the motions in both switch regions.

Considering that RACB bound to GTP-analogs exhibits pronounced line broadening in the NMR spectra, we hypothesize that slow dynamics (μs-ms range) might be the defining feature of the activated state. To

quantify motions on this timescale, we conducted $^{15}$N-CPMG relaxation dispersion experiments[49] (Fig. 3a). In inactive RACB-GDP (top panel in Fig. 3a), only a low number ($n = 9$) of backbone amides show μs-ms dynamics with exchange rates, $k_{ex}$, of 100–1500 Hz (Fig. 3b, red symbols). Some of these residues are in the switch regions (T30, T33 and R76) but also in loop regions that are not involved in conformational switching (Fig. 3d). In contrast, activated RACB-GMPPNP shows a higher number of residues with μs-ms motions (Fig. 3a, lower panel, Fig. 3f) in functionally important regions, including the P-loop (V12, T20), switch I (N29, T33) and II (D66, S74) and to a lower extent in the insert region. Strikingly, a phosphomimetic mutation in switch II (S74E) has been reported to affect the signaling properties of ROP proteins, highlighting the functional significance of this region[50]. In addition, large parts of the wider nucleotide-binding pocket exhibit exchange motions, indicating that the plasticity of RACB is enhanced in the activated state. This is evident by the higher number of backbone amides showing exchange motions ($n = 29$), and by the

**Fig. 4 | Nucleotide-dependent interaction between RACB and RIPb. a** Isothermal titration calorimetry (ITC) with RIPb-CC2 (yellow cartoon) and RACB-GDP (left, red cartoon) or RACB-GTPγS (right, blue cartoon). No binding was observed for the GDP-bound form, while RACB-GTPγS exhibited binding with a dissociation constant ($K_D$) of 1.7 ± 0.6 μM. Values denote the mean value and S.D. obtained from $n$ = 3 technical replicates. **b** Difference in deuterium uptake of active RACB-GTPγS upon RIPb-CC2 binding after 10 s exchange time. Positive values (red bars) indicate increased protection in the complex. Gray bars represent minor changes. Gray shadings indicate the nucleotide binding sites and switch regions. Peptide coverage was 89% (Supplementary Fig. 14a). **c** Relative fractional deuterium uptake mapped on the RACB-GTPγS crystal structure. **d** HDX effects mapped on the sequence of RIPb-CC2 after complex formation with RACB-GTPγS after 10 s exchange time. Positive values (red bars) identify the RIPb-CC2 binding site, which includes a cluster of residues (>10% deuterium uptake) aligning with the QWRKAA motif (red box at the top). Peptide coverage was 91% (Supplementary Fig. 14b).

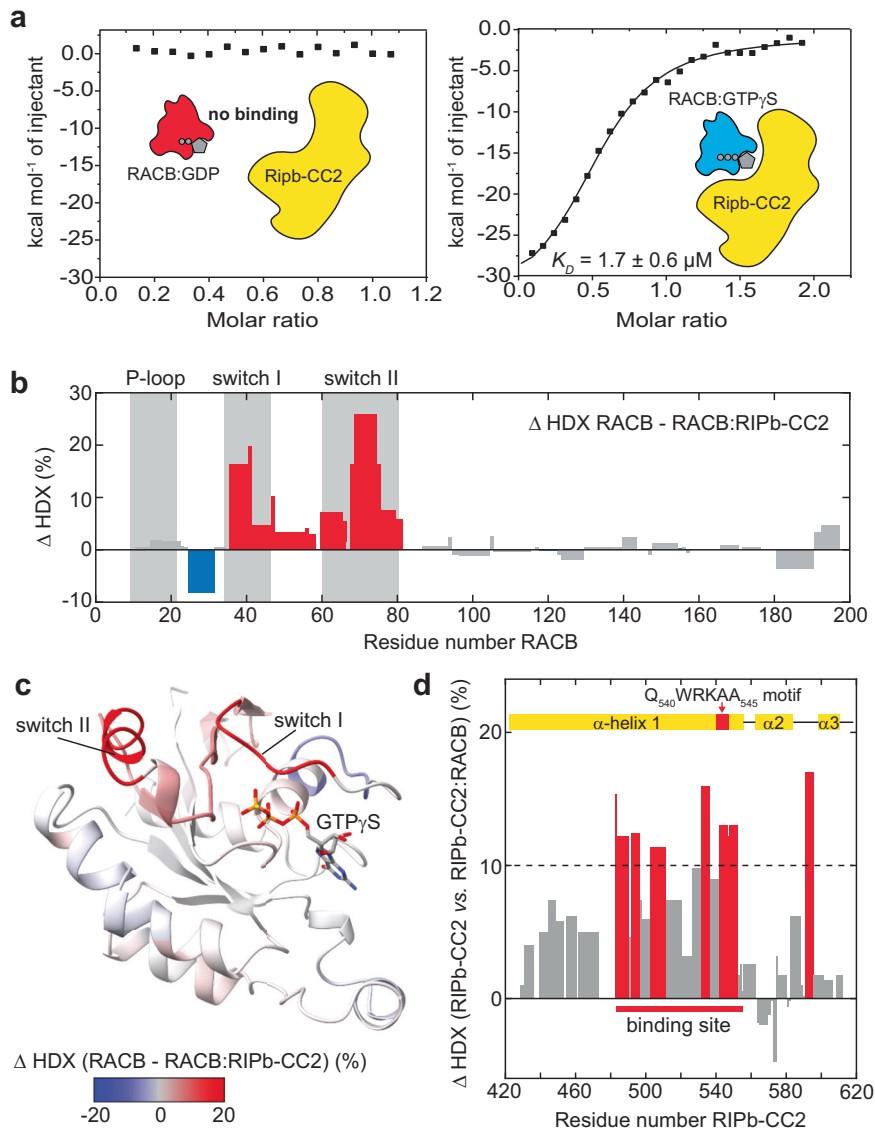

significantly higher average exchange rates in RACB-GMPPNP (Fig. 3b, blue symbols). The higher degree of conformational exchange in RACB-GMPPNP is exemplarily shown for Asn29 in switch I and Asp66 in switch II (Fig. 3c, d) where no exchange could be detected in the GDP-bound form, but clear NMR relaxation dispersion curves were observed with RACB-GMPPNP.

## GTP-bound active RACB interacts with the downstream effector RIPb

Next, we conducted biophysical binding assays with the effector protein RIPb, which is required for membrane remodeling during fungal attack[41]. The C-terminal coiled-coil domain of RIPb (RIPb-CC2) was previously shown to be the interaction site with RACB[41]. To probe the interaction by biophysical methods, we first recorded 2D-[$^{15}$N,$^1$H]-TROSY NMR experiments of RACB-GDP or RACB-GTPγS upon the addition of RIPb-CC2 (Supplementary Fig. 9a, b). While no spectral changes were observed with RACB-GDP, RACB-GTPγS displayed pronounced line broadening effects, resulting in the disappearance of most resonances in well-ordered regions. These effects indicate a nucleotide-dependent interaction of RIPb-CC2 with the activated form of RACB, but not with the inactive GDP-bound state. This behavior is consistent with the canonical functional model of a small GTPase, acting as a G-nucleotide-dependent switch for downstream signaling[51].

To further quantify the binding effects detected by NMR, we performed isothermal titration calorimetry (ITC) experiments with RACB bound to GDP or GTPγS, and RIPb-CC2 (Fig. 4a, S9c). While no measurable binding was detected with RACB-GDP (left panel), a well-defined binding curve with a dissociation constant ($K_D$) of 1.7 ± 0.6 μM was obtained in the presence of GTPγS. Furthermore, CD data and ITC binding experiments at different temperatures indicate that the α-helical coiled-coil domain of RIPb-CC2 is not fully structured at 20 °C (Supplementary Fig. 10a) but folds upon binding to RACB, giving rise to a high binding enthalpy at 20 or 25 °C but not at 15 °C (Supplementary Fig. 10b, c). To evaluate nucleotide-dependent changes in the active state of RACB, we further determined a $K_D$ value between RACB-GMPPNP and RIPb-CC2 by ITC (Supplementary Fig. 11a), which is with 15 μM about 8-fold weaker than with GTPγS. To validate these numbers derived from ITC, we also determined $K_D$ values by microscale thermophoresis (MST) using RED-NHS-labeled RACB (Supplementary Fig. 11b, c). Consistent with the ITC results, these measurements revealed that RACB-GTPγS has a 4 to 5-fold higher affinity for RIPb-CC2 than RACB-GMPPNP ($K_D$ of 76 *versus* 330 nM). The slightly higher affinity obtained by MST as compared to ITC can be rationalized by the presence of a hydrophobic fluorescent dye attached to RACB. To further validate the $K_D$ values from the biophysical experiments, we next performed SEC and chemical crosslinking experiments with RACB and RIPb-CC2. In SEC, RACB-GTPγS or RACB-

**Fig. 5 | Structure of the RACB-RIPb effector complex. a** Only minimal structural changes as seen in the overlay of the crystal structures of RACB-GTPγS alone (purple) and in the complex with RIPb (gray) highlight that RACB-GTPγS populates a fully active conformation. **b** Crystal structure of RACB-GTPγS bound to the C-terminal coiled-coil domain of RIPb (RIPb-CC2). **c** Details of the RACB-RIPb complex showing the binding interface between the switch regions of RACB and the conserved QWRKAA motif of RIPb. **d** RIPb Q540L/W541G does not bind to RACB-GTPγS as detected by microscale thermophoresis (MST). Error bars were derived from three technical replicates. Individual data points are shown as gray symbols, mean values and S.D. as black symbols and error bars.

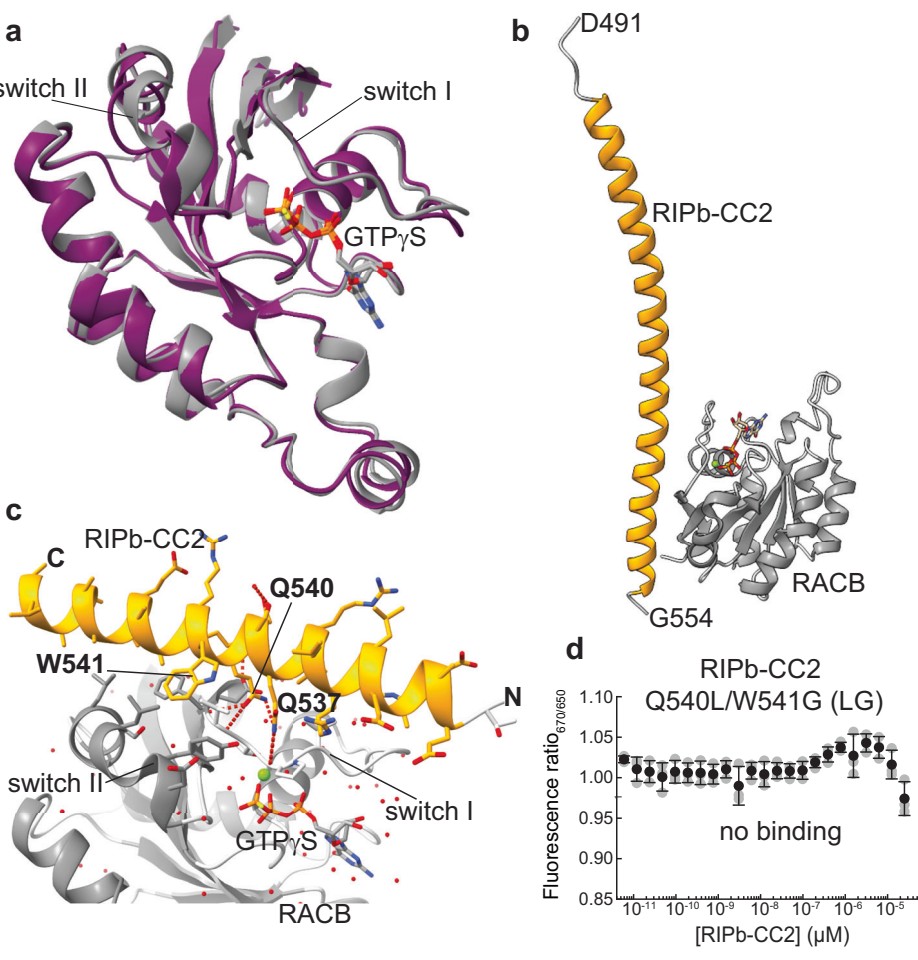

GMPPNP and RIPb-CC2 did not co-elute in a single peak corresponding to the complex (Supplementary Fig. 12a, b), which is arguing for the μM affinity obtained by ITC. In chemical crosslinking assays with the amino-selective crosslinker BS[3] (Supplementary Fig. 12c) the dimerization tendency of RIPb-CC2 could be confirmed by the presence of a strong dimer band, together with the nucleotide-dependent interaction between RACB and RIPb-CC2, where only GTPγS and GMPPNP gave rise to higher molecular weight bands, representing the complex.

Sequence analyses of ICRs/RIPs from different plants showed that the C-terminus of these proteins contains conserved regions, specifically a motif called QWRKAA[40], which is also found in barley RIPb. This motif has been suggested to be the primary binding site for ROP GTPases. Furthermore, the C-terminus of RIPs was deemed essential for their proper joint localization with ROPs at the plasma membrane, whereas it is dispensable for ICR/RIP-microtubule association[40–42]. However, a structural picture of such an ROP-RIP complex remained elusive so far. To explore the structural features of the RACB-RIPb-CC2 complex, we first used HDX-MS[52,53] to obtain information on the interaction sites on both proteins (Supplementary Fig. 13). While NMR titrations and ITC confirmed a specific, high-affinity interaction between RIPb-CC2 and RACB-GTPγS, HDX-MS enabled us to identify the regions in both proteins involved in complex formation. Mapping the differences in deuterium uptake of RACB *versus* RACB in complex with RIPb-CC2 onto the RACB sequence (Fig. 4b) revealed pronounced exchange protection in switch I (residues 35–41) and the highest protection (~25%) in switch II (residues 68–75), revealing that these regions are the main interaction site for RIPb (Fig. 4c).

The involvement of switches I and II also explains the strong nucleotide-dependence in the complex formation. Similarly, the HDX profile of RIPb-CC2 showed reduced H/D exchange in the complex with RACB, mainly in the C-terminal half (residues 480–550) of the first

predicted longer α-helix that harbors the QWRKAA motif (Fig. 4d), corroborating that this region might be the primary binding site for RACB.

## High-resolution structure of the RACB-RIPb complex

We next set out to determine a high-resolution X-ray structure of the RACB:GTPγS-RIPb-CC2 complex. Despite intense crystal screening attempts, only poorly diffracting crystals were initially obtained, presumably due to high flexibility in RIPb-CC2. Guided by our HDX-MS data (Fig. 4d), we designed a shorter RIPb-CC2 construct, named RIPb-CC2-C5, encompassing residues 485–555, which included the $Q_{540}WRKAA_{545}$ motif in the putative coiled-coil helical region but lacked the predicted unfolded C-terminal tail.

This optimized RIPb construct finally resulted in high-quality crystals and a high-resolution structure of the complex. The RACB-GTPγS structure in the complex with RIPb represents the fully active state, which is very similar to RACB-GTPγS alone. In both structures, the switch regions are present in a more closed conformation with the nucleotide tightly sandwiched. However, in the complex with RIPb, switch II of RACB adopts a higher degree of helical secondary structure (Table 1, Fig. 5a). The almost identical structures of RACB:GTPγS and the RACB:GTPγS-RIPb-CC2 complex suggests that GTPγS alone can induce the fully active and effector-protein-binding-competent state (Fig. 5a). The RACB:GTPγS-RIPb-CC2 complex crystallized in two distinct forms. In both cases the asymmetric unit of the crystals contains one copy of RACB interacting with one copy of the RIPb-CC2 construct (Supplementary Fig. 15). In the first crystal form, resolved at a resolution of 2.3 Å, a longer fragment of the RIPb-CC2 region is visible with RACB interacting at the C-terminal end of the helical region (Fig. 5b) with an anti-parallel orientation of two RIPb-CC2 monomers (Supplementary Fig. 15a) in the crystal lattice. In the second crystal form, which was resolved at higher resolution (2.07 Å), only the core interaction

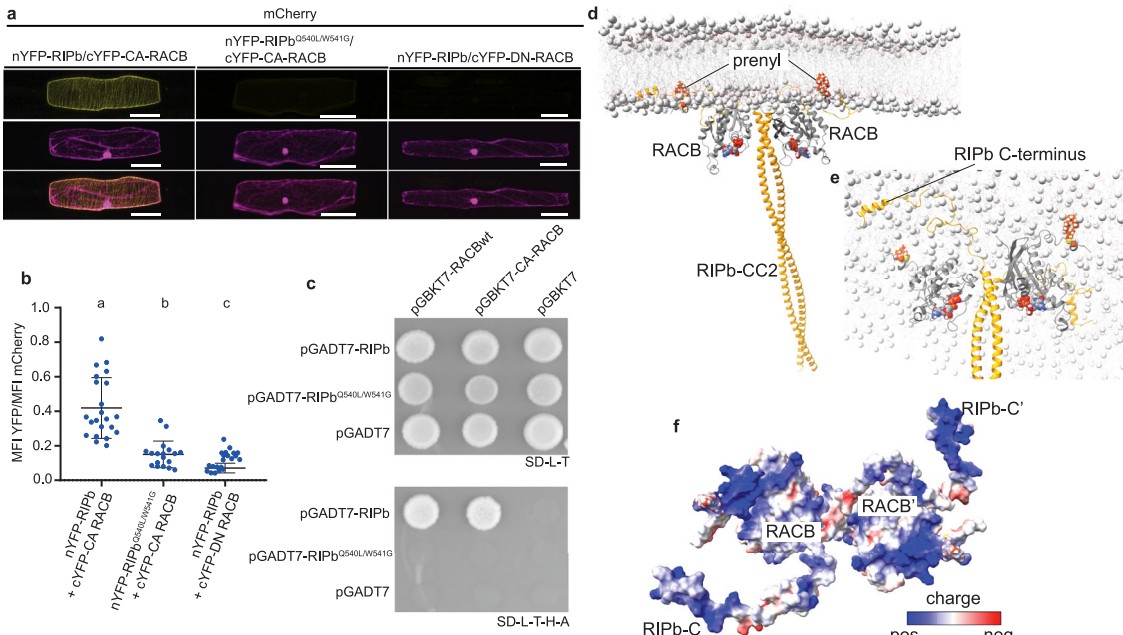

**Fig. 6 | Interaction, subcellular localization and functional model of RACB and RIPb in planta. a** Bimolecular fluorescence complementation (BiFC) assays with single epidermal cells transiently transformed with split-YFP constructs. Images represent typical cell recordings of a minimum of 20 cells per experiment and from two independent transformation experiments with similar results. Images represent z-stacks of 25 confocal sections, each with a 1.5–2 μm increment. Scale bar = 50 μm. **b** Quantification of BiFC signals from images was performed with constant settings. Signal intensity (mean fluorescence intensity [MFI]) was measured over the whole cell. The ratio between split-YFP and free mCherry signal was calculated. Signals were measured in 10 cells for each construct. Letters indicate significance by one-way ANOVA (Tukey's multiple comparison test; $P = 0.05$). **c** Yeast-two-hybrid assays

with wild-type RACB and CA RACB in bait plasmids and RIPb constructs in prey plasmids. As negative control, empty vectors (EV) were used. For transformation control, yeast was dropped on SD medium without leucine (-Leu) and tryptophan (-Trp). For the selection of positive interactions, SD medium was used additionally lacking adenine (-Ade) and histidine (-His). **d** Model of prenylated dimeric RACB-RIPb at the membrane. RACB is anchored through its C-terminal prenylated cysteine 194. RIPb is depicted in yellow, RACB in gray. **e** View from top down the membrane. The C-terminus of RIPb (yellow) binds to the membrane surface. The gray spheres in the membrane represent the phosphate moieties. **f** Electrostatic surface coloring of the RACB-RIPb complex indicates a positive charge on RACB and RIPb, facilitating membrane attachment.

region of RIPb is visible (Fig. 5c) but with a parallel orientation of RIPb-CC2 in the dimer observed in the crystal lattice (Supplementary Fig. 15b, Table 1). Despite different arrangements the interaction between the RACB and RIPb proteins was identical in both crystal forms. The higher-resolution complex structure showed better quality and was therefore used for further analysis. The core of the interaction is established by the RIPb side chain amides of Q537 and Q540 forming hydrogen bonds to the backbone carbonyl of P37 and amides of V39 and F40, respectively, located in switch I of RACB, as well as by RIPb W541 which is involved in interactions with hydrophobic and aromatic side chains of switch I and II in RACB. Strikingly, two of these residues in RIPb are in the highly conserved $Q_{540}$WRKAA$_{545}$ motif. To validate the complex structure, we introduced a double mutation in RIPb-CC2 to yield Q540L/W541G-RIPb-CC2 and performed an MST binding assay. In contrast to the wild-type protein (Supplementary Fig. 11), the variant has lost its binding affinity (Fig. 5d), supporting the conclusion that the QWRKAA motif is the primary interaction site for RACB.

**Validation of the RIPb-RACB interaction in barley**

In *A. thaliana,* it has been shown that the QWRKAA motif in RIP proteins is necessary for ROP interaction[39]. To validate the structural and in vitro binding results (Fig. 4, 5) as well as the importance of the QWRKAA motif in the barley system studied herein, we conducted *in planta* bimolecular fluorescence complementation (BiFC) assays, where the fluorescent protein YFP was assembled from fragments that are fused to interacting protein partners, giving rise to yellow fluorescence (Fig. 6a). Co-expression of RIPb and CA (constitutively activated, G15V[41]) RACB in barley cells resulted in strong YFP fluorescence, confirming that YFP reconstitution takes place. In contrast, the Q540L/W541G RIPb variant failed to restore the fluorescence signal in presence of CA RACB (Fig. 6a). As a negative control, RIPb was

used together with dominant-negative RACB (DN RACB, T20N[41]) that does not adopt an active, binding-competent state, verifying the specificity of the interaction (Fig. 6a). Ratiometric quantification of the YFP-specific compared to co-expressed red fluorescing mCherry signal confirmed these observations and showed a significant decrease in YFP fluorescence for CA RACB-RIPb-Q540L/W541G compared to CA RACB-RIPb wild-type (Fig. 6b). The lower level with the RIPb variant harboring two mutations in the QWRKAA motif is almost identical to the binding incompetent DN-RACB. Importantly, the highest yellow fluorescence signal was located at the cell periphery and microtubules, consistent with the previously reported localization of RIPb and the proposed function of the RIPb-RACB complex in membrane remodeling during fungal invasion[41], as well as its interaction with phospholipids in vitro[29]. To validate this interaction in an independent system, we performed yeast two-hybrid (Y2H) assays (Fig. 6c) using wild type or Q540L/W541G full-length RIPb constructs. Wild-type RIPb interacted with RACB wild-type or CA RACB[41] as indicated by the observation of yeast cell growth under interaction-selective conditions (lower panel in Fig. 6c). As expected from a loss in the interaction, RIPb Q540L/W541G completely prevented yeast growth when combined with both wild-type RACB and CA RACB (Fig. 6c), identical to the behavior with the empty control plasmids. These findings demonstrate that the QWRKAA motif is an essential interaction site in RIPb for RACB, and that in barley cells this interaction takes place at the cell membrane and cortical microtubules, consistent with its proposed function[41].

To better visualize how such a RACB-RIPb complex can be attached to the plasma membrane and considering the dimeric nature of RIPb (Supplementary Figs. 2, 11), we constructed a dimeric (2:2) structural model in which RACB carries a natively occurring C-terminal farnesyl moiety at Cys194. To complete the missing structural coordinates at the N-terminal

end of RIPb-CC2, we used an AlphaFold model and structurally aligned the experimental complex structure (see "Methods"). Using previous protocols[54,55], the assembled complex was in silico attached to a POPC/POPG (50:50%) bilayer membrane and subjected to a molecular dynamics simulation to optimize the interaction between the protein complex and the membrane. The resulting model showed that the farnesyl moieties of both RACBs are inserted into the bilayer membrane, mediating stable membrane-anchoring of the complex (Fig. 6d). In addition, the C-terminal tails of RIPb interacted with the membrane surface (Fig. 6e). The attachment of the complex to the membrane surface is further facilitated by the positively charged and membrane-facing surfaces of RACB and RIPb, including the positively charged C-terminal regions of RIPb (Fig. 6f). In full-length RIPb, the membrane binding pose is expected to be identical since the missing N-terminal portions of RIPb protrude away from the membrane surface. Taken together, our structural and cellular data provide strong evidence for a stable RACB-RIPb complex at the plasma membrane, serving as an anchor point for downstream ROP signaling.

## Discussion

To date, only a few isolated ROPs in GDP- or GTP-analog-bound states have been structurally characterized. These include GDP-bound *Arabidopsis thaliana* ROP9[18], the PRONE-GEF-bound structures of ROP4 and ROP7[18–20], and rice GMPPNP-bound RAC1[17]. The available structures consistently show the conserved Ras-fold with canonical G-box motifs, a Rho-specific insert helix and the switch I and II regions that undergo conformational changes upon nucleotide exchange[44]. Moreover, no high-resolution structure of a plant ROP in complex with a downstream effector has been reported so far, leaving the question unresolved on how ROPs undergo nucleotide-dependent activation and engage with plant-specific effector proteins for downstream signaling.

Our study provides high-resolution structures of barley RACB in its GDP-, GMPPNP- and GTPγS-bound states, as well as in complex with the ROP effector RIPb (Figs. 1, 5, 6). These structures delineate the complete activation trajectory of a plant ROP and reveal how RACB progresses from an inactive to a pre-active and finally to a fully active conformation. Despite only 55–65% sequence identity to mammalian RAC1 proteins[8,28,30], RACB undergoes a conserved series of switch-region rearrangements, including γ-phosphate stabilization and formation of an effector-competent interface. Consistent with extensive studies on mammalian Rho GTPases (reviewed, e.g., in ref. 56), NMR and HDX-MS reveal that RACB exhibits μs-ms conformational dynamics rather than sampling just a binary on/off equilibrium (Fig. 2, 3). This intrinsic dynamic landscape governs effector accessibility and is stabilized in the fully active state upon binding to RIPb.

A comparison of the structural features of the nucleotide states supports a two-step activation model analogous to mammalian Rho proteins[57,58]. RACB-GMPPNP adopts a pre-active form with an open switch I that facilitates nucleotide exchange but confers weak effector binding, whereas RACB-GTPγS stabilizes a fully active conformation competent for high-affinity RIPb interaction. Our RACB-RIPb complex structure provides an atomic description of effector engagement in plant ROP signaling. Binding specificity arises from the conserved QWRKAA motif in RIPb where the glutamine residue forms stabilizing polar interactions and the conserved tryptophan residue inserts deeply between switch I and II (Fig. 5). This mode of recognition parallels mammalian RhoA-ROCKI interactions involved in tumor invasion, where a short helix in ROCKI inserts hydrophobic residues into an active-state cleft to stabilize the GTP-bound conformation[59].

The RACB-RIPb interaction is intimately linked to membrane association and has been suggested to execute membrane remodeling during fungal invasion. Assembly of the RACB-RIPb complex at the membrane surface using computational methods (Fig. 6) indicates that RACB's prenylated C-terminus and basic surface residues of both proteins mediate cooperative membrane engagement, positioning the complex at the cytosolic leaflet-consistent with their previously observed accumulation at the cell membrane and fungal penetration sites[41]. Interaction of RACB and full-

length RIPb establishes a connection between the plasma membrane and cortical microtubules[41] (Fig. 6). This spatial organization provides a possible mechanistic explanation for how active RACB and RIPb bridge cytoskeletal and membrane remodeling during the susceptibility responses. Analogous to *Arabidopsis thaliana* ROP signaling, where ICR/RIP effectors modulate actin and microtubule dynamics by forming membrane nanodomains with active ROPs[60,61], the RACB-RIPb complex likely serves as a susceptibility module that integrates ROP activation at the cell membrane with local reorganization of the cytoskeleton and membrane dynamics required for fungal invasion[34,41]. Along with the cytoskeleton and GAP- and GEF-regulated ROPs, ICR/RIP proteins can significantly change membrane heterogeneity and lead to cellular symmetry breaking[62,63].

Future studies will show how ROP-RIP-membrane association influences regulatory interactions between the plasma membrane and the cytoskeleton. Plasma membrane lipid composition likely contributes to this behavior, as ROPs, including RACB, bind anionic phospholipids and rely on their polybasic C-terminus for proper signaling function[24,29,64]. Our structure-based modeling (Fig. 6) suggests that ROPs, together with RIPs, form a positively charged interface that interact with negatively charged phospholipids. Interestingly, barley RIC proteins (ROP-Interactive and CRIB-(Cdc42/Rac Interactive Binding) motif-containing), another class of ROP effectors, also contain polybasic protein domains and have also been reported to interact with RACB at the plasma membrane[65]. This raises the possibility that ROPs and ROP effectors more generally use polybasic protein stretches in protein-protein and lipid interactions.

Our findings demonstrate that RACB function is governed by the dynamic state of its switch regions, which modulate effector binding in a spatiotemporally precise manner (Fig. 3 & Supplementary Fig. 8). These results establish that key principles of GTPase regulation - including dynamic switching, conformational selection, and effector-induced stabilization - are evolutionarily conserved across eukaryotes. Yet, RACB exhibits plant-specific features, including a C-terminal HVR motif for membrane localization[29] and a unique regulatory system involving ROP-GEFs, CRIB-containing ROP-GAPs, and selective effector proteins[8,44]. Thus, a finely tuned plasticity of the switch-regions is a key factor for the regulation of ROP-effector engagement at the plasma membrane.

## Methods
### Construct design and cloning
RACBwt (Uniprot ID Q8RW50) and RIPb-CC2 (421-612) (for amino acid sequences, see Supplementary Note 1) from barley (*Hordeum vulgare*) DNA coding sequences were cloned into the *E. coli* expression vector pET21a by restriction-free cloning[66], containing an N-terminal hexa-histidine tag, followed by a GB1 fusion protein and a thrombin cleavage site. RACB-ΔC (1-179) and RIPb-CC2 variants, RIPb-CC2-C5 (485-555) and RIPb-CC2 (Q540L/W541G) were generated by site-directed mutagenesis using the QuikChange Lightning Site-Directed Mutagenesis Kit (Agilent).

### Protein production
All plant protein constructs were expressed in *E. coli* BL21 (DE3) cells. Cells were grown in LB media with 100 μg/ml ampicillin at 37 °C until $OD_{600}$ reached 0.6-0.8. For producing $^{15}N$ labeled RACB, cells were grown in isotope-enriched M9 media containing 1 g/L $^{15}N$ ammonium chloride and 2 g/L glucose, while 1 g/L $^{15}N$ ammonium chloride and 2 g/L $^{13}C$ glucose were used to produce $^{13}C,^{15}N$-labeled protein for 3D triple resonance experiments. RACB protein expression was induced with 0.2 mM IPTG, and cells were grown at 20 °C for 18–20 h. RIPb-CC2 constructs were induced with 0.5 mM IPTG, followed by incubation at 37 °C for 4 h. Cells were harvested by centrifugation at 5000 × g for 20 min at 4 °C, flash-frozen in liquid nitrogen, and stored at –80 °C. For RACB purification, cell pellets were lysed in a buffer containing 20 mM HEPES pH 7.5, 200 mM NaCl, 5 mM $MgCl_2$, 10% glycerol, 10 mM β-mercaptoethanol (β-ME), 10 mg/ml lysozyme, one tablet of Complete Protease Inhibitor Cocktail (Roche), DN*ase*I (50 μg/ml) and 50 μM GDP. The lysate was centrifuged at 18000 rpm for 30 min, and the supernatant was applied to a Ni-NTA

column equilibrated with lysis buffer. The column was then washed with lysis buffer, followed by a wash buffer containing 20 mM HEPES pH 7.5, 500 mM NaCl, 5 mM MgCl$_2$, 10% glycerol, 10 mM β-ME and 10 mM imidazole. Proteins were eluted using an elution buffer with 20 mM HEPES pH 7.5, 200 mM NaCl, 5 mM MgCl$_2$, 10% glycerol, 10 mM β-ME, 300 mM Imidazole and 50 μM GDP.

RIPb-CC2 and RIPb-CC2 (Q540L/W541G) were purified under denaturing conditions by adding 6 M guanidine hydrochloride (GuHCl) while lysing the cells in the same buffer used for RACB purification, without the addition of GDP. This was followed by a gradual decrease in GuHCl concentration in the wash buffers from 6 M to 3 M and then to 0 M, with the protein eluted using an elution buffer containing 20 mM HEPES pH 7.5, 200 mM NaCl, 10% glycerol and 300 mM Imidazole. RACB and RIPb-CC2 variants were further purified by removing the His$_6$-GB1 tag with thrombin, followed by final purification through size-exclusion chromatography (SEC) using a Superdex™ 75 HiLoad 16/600 column (Cytiva) in SEC buffer (25 mM HEPES pH 7.5, 100 mM NaCl, 5 mM MgCl$_2$ and 2 mM DTT). Fractions containing the desired proteins were concentrated using Amicon Ultra-10 centrifugal devices with a 10 kDa MWCO to a final concentration of between 300 and 500 μM. For RACB, additional GDP (50 μM) was added. Samples were subsequently flash-frozen and stored at –80 °C until further use.

### Nucleotide exchange
To prepare GTPγS- or GMPPNP-bound RACB[67], purified RACB-GDP was buffer exchanged to 20 mM Tris pH 7.5, 100 mM NaCl, 5 mM MgCl$_2$, and 5 mM DTT, then concentrated to a concentration of 500 μM in a final volume of 0.5 ml. Protein concentration was determined using a nanodrop by assessing absorption at 280 nm, considering the extinction coefficient of the protein and the nucleotide (ε ≈ 7720 M$^{-1}$·cm$^{-1}$). To initiate nucleotide exchange, GTPγS or GMPPNP (Jena Bioscience) was added at a tenfold molar excess relative to the protein concentration. Subsequently, alkaline phosphatase (Roche) was added at a concentration of 2 U/mg of protein. The sample was adjusted to 1.5 mL with nucleotide exchange (NE) buffer, consisting of 40 mM Tris pH 7.5, 200 mM (NH$_4$)$_2$SO$_4$, 10 μM ZnCl$_2$, and 5 mM DTT, followed by incubation at 4 °C on a rolling device overnight. The reaction was stopped by applying the sample to a size-exclusion chromatography (SEC) column Superdex™ 75 Increase 10/300 GL in 20 mM HEPES pH 7.5, 100 mM NaCl, 1 mM MgCl$_2$, and 2 mM DTT. The final sample was supplemented with an additional 10-fold molar excess of the desired nucleotide for crystallization.

### Circular dichroism (CD) spectroscopy
All CD measurements were performed using a Jasco J-1500 spectrophotometer with a 1 mm path length quartz cuvette and a bandwidth of 1 nm in CD buffer containing 20 mM NaP$_i$ 7.0, 50 mM NaCl, and 5 mM MgCl$_2$. CD spectra were recorded at 5 °C or 20 °C, respectively, with a concentration of approximately 10 μM, covering wavelengths from 260 nm to 190 nm, with a total of five accumulations at a scanning speed of 100 nm/min. CD thermal unfolding was measured from 20 to 100 °C at the respective wavelengths with a heating rate of 1°/min.

### Crystallization, diffraction, data collection and processing
For crystallization, RACB-GDP, RACB-GMPPNP, and RACB-GTPγS were concentrated to 13 mg/mL. To form the complex, RACB-GTPγS was mixed with the RIPb-CC2-C5 construct in equal molar ratio at a final concentration of 10 mg/mL and incubated for 30 min at room temperature. The crystallization experiments for RACB complexes were conducted at the X-ray Crystallography Platform at Helmholtz Zentrum München. The initial screening for all variants was performed at 292 K using a Nanodrop dispenser in sitting-drop 96-well plates and commercial screens. After selecting the best hits from the screening, manual optimization was carried out. The best X-ray diffraction data set for RACB-GDP was obtained from a crystal grown with 180 mM Mg formate, 100 mM Na acetate pH 4.0, and 17% (w/v) PEG 5000 MME. The RACB-GMPPNP complex produced the best diffraction from a crystal formed with 300 mM NaF and 18% (w/v)

PEG 3350. The crystals for RACB-GTPγS which diffracted the best grew from the solution composed of 18% (w/v) PEG 3350 and 200 mM Na formate. The best diffracting crystals for RACB-GTPγS-RIPb-CC2-C5 (parallel) and RACB-GTPγS-RIPb-CC2-C5 (antiparallel) grew from 20% (w/v) PEG 6000, 100 mM HEPES pH 7.0, 200 mM NaCl, and 10 mM MgCl$_2$, and 50 mM MES pH 5.6, 1.8 M Li$_2$SO$_4$, respectively. For the X-ray diffraction experiments, the crystals were mounted in nylon fiber loops and flash-cooled to 100 K in liquid nitrogen. Cryoprotection was performed for 5 seconds in reservoir solution supplemented with 25% (v/v) ethylene glycol. All X-ray diffraction data were collected at the P11 beamline (PETRA III, DESY, Hamburg). Data collection was carried out at 100 K. The data sets were indexed and integrated using XDS[68] and scaled with SCALA[69,70]. Intensities were converted to structure-factor amplitudes using the program TRUNCATE[71]. Table 1 summarizes data collection and processing statistics for all RACB complexes.

### Structure determination and refinement
The structure of the RACB-GDP complex was determined by molecular replacement (MolRep)[72] with the Rac-like GTP-binding protein *At*RAC5 from *Arabidopsis thaliana* (PDB ID: 2NTY)[19] as a search model. The resulting RACB-GDP structure was subsequently used as a search model to solve the structures of the remaining complexes. The RIPb-CC2 helix was built manually. Model building and rebuilding for all variants were carried out using COOT[73]. Refinement was performed with REFMAC5[74] using a maximum-likelihood target function for RACB-GDP, RACB-GMPPNP, and parallel RACB-GTPγS-RIPb-CC2 complex, and with PHENIX[75] for RACB-GTPγS and antiparallel RACB-GTPγS-RIPb-CC2 complex. The stereochemical quality of the final models was assessed with PROCHECK[76] and MolProbity[77]. The Ramachandran plot statistics indicate overall good stereochemical quality for all refined models. For the RACB-GDP structure (PDB ID 9T3C), 98% of residues fall within the most favored regions of the Ramachandran plot, with the remaining 2% located in additionally allowed regions. The RACB-GMPPNP structure (PDB ID 9T3D) shows excellent geometry, with 100% of residues in the most favored regions and none in additionally allowed regions.

For the RACB-GTPγS structure (PDB ID 28NN), 93% of residues are found in the most favored regions and 6% in additionally allowed regions. In the RACB-GTPγS-RIPb-CC2 parallel complex (PDB ID 9T3F), 96% of residues occupy the most favored regions, while 4% fall within additionally allowed regions. The RACB-GTPγS-RIPb-CC2 antiparallel complex (PDB ID 9T3E) shows 87% of residues in the most favored regions and 9% in additionally allowed regions. Final refinement statistics for all models are shown in Table 1.

### NMR spectroscopy and backbone resonance assignment
NMR experiments were conducted at 25 °C or 30 °C using Bruker Avance III HD NMR spectrometers operating at $^1$H frequencies of 600, 800 and 950 MHz equipped with cryogenic TXI probes. 2D-[$^{15}$N, $^1$H]-HSQC spectra were acquired with 0.1 mM uniformly $^{15}$N-labeled RACB at 25 °C. For three-dimensional triple resonance experiments, the RACB concentration was increased to 0.8 mM in NMR buffer (20 mM NaP$_i$ pH 7.0, 50 mM NaCl, 1 mM MgCl$_2$ and 2 mM DTT) containing 7% (v/v) D$_2$O and 2 mM GDP or GMPPNP, respectively. Backbone resonance assignments of wild-type RACB (1–197) bound to GDP or GMPPNP were obtained using a suite of five triple resonance experiments[45], including 3D-HNCA, HNCO, HN(CO)CA, HN(CA)CO, HNCACB, as well as a 3D-$^{15}$N-edited [$^1$H,$^1$H]-NOESY experiment (150 ms mixing time). NMR data were processed using Topspin4.0 (Bruker Biospin) and analyzed with NMRFAM-Sparky[78].

Heteronuclear {$^1$H},$^{15}$N-NOE experiments were measured at 950 MHz $^1$H frequency with 0.5 mM $^{15}$N-labeled RACB-ΔC (1-179) for both GDP- and GMPPNP-bound forms at 30 °C. The reported het. NOE values were derived from the peak intensity ratio of resolved amide cross peaks in the $^1$H-$^{15}$N NMR correlation spectrum after 2.5 s $^1$H pre-saturation and without $^1$H saturation. Chemical shift perturbation experiments were calculated using the empirical formula $\Delta d_{av} = [(\delta \Delta H^2 + (\delta \Delta N/5)^2)/2]^{0.5}$ [79].

2D single quantum [15]N-CPMG relaxation dispersion experiments[49] were recorded at 600 and 950 MHz [1]H frequency in a pseudo-3D manner with a 20 ms constant-time [15]N relaxation delay with $\tau_{cp}$ spacing corresponding to 16 different CPMG-based RF field strengths, $\nu$(CPMG), ranging from 25 to 1000 Hz, including a reference experiment without the 20 ms relaxation block and three duplicates (at low, medium and high CPMG RF field strengths) for error extrapolation. All spectra were recorded at 30 °C with 0.5 mM [15]N-labeled RACB-$\Delta$C (1-179) after addition of 5 mM GDP or 5 mM GMPPNP. 128 complex data points were recorded in the indirect dimension with 64 scans per increment. Data analysis was done with the program Relax[80]. Initial starting values were determined by grid search and data subsequently fitted to motional models no exchange, fast exchange[81] or a general two-site exchange model[82] for all time scales. Each spin system was assigned to the best-fitting model based on Chi[2] calculations. Finally, the statistical significance of each model was estimated with Monte-Carlo simulations.

### Hydrogen-deuterium exchange (HDX) mass spectrometry

Hydrogen-deuterium exchange (HDX) analyses were performed on an ACQUITY UPLC M-class system with automated HDX technology (Waters, Milford, MA, USA). Experiments were conducted at 20 °C with sampling at 0, 10, 60, 600, 1800, and 7200 s, each in technical duplicate. For each time point, 3 µl of approximately 30 µM protein solution was diluted 1:20 in 99.9% $D_2O$-based 20 mM sodium phosphate buffer, pH 6.8 (adjusted with HCl), or an equivalent $H_2O$-containing reference buffer. The reaction was quenched by mixing 1:1 with 200 mM $KH_2PO_4$ and 200 mM $Na_2HPO_4$, pH 2.3 (adjusted with HCl), supplemented with 4 M guanidine hydrochloride and 200 mM TCEP at 1 °C. Samples (50 µl) underwent on-column peptic digestion using a Waters Enzymate BEH pepsin column (2.1 × 30 mm) at 20 °C. Peptides were separated via reverse-phase chromatography at 0 °C, with a gradient increasing acetonitrile from 5–35% over 6 min, then 35–40% in the next minute, and 40–95% in the following minute. A Waters Acquity UPLC C18 1.7 µm Vangard 2.1 × 5 mm trapping column and a Waters Acquity UPLC BEH C18 1.7 µm 1 × 100 mm analytical column were used. Eluted peptides were detected and characterized using an in-line Synapt G2-S QTOF HDMS mass spectrometer (Waters, Milford, MA, USA). All measurements were performed in biological duplicate, with MS data collected across an m/z range of 100–2000. Calibration with Glu-fibrino peptide B (Waters, Milford, MA, USA) ensured mass accuracy. Peptide identification was done through MSE, with collision energy automatically ramped from 20–50 V. Since the automated platform maintains consistent sample conditions, back exchange corrections were not applied, and deuterium incorporation is reported as relative levels (Wales and Engen, 2006). Data analysis was carried out using PLGS 3.0.3 and DynamX 3.0 (Waters, Milford, MA, USA).

### Isothermal calorimetry (ITC)

Isothermal titration calorimetry (ITC) measurements were performed at 20 °C using a MicroCal PEAQ-ITC instrument (Malvern Panalytical). 200 µM RACB-GDP was exchanged to RACB-GTPγS or GMPPNP by addition of 10 mM EDTA and incubating for 10 min at room temperature. Subsequently, 2 mM GTPγS or GMPPNP was added, and the sample was incubated on ice for 2 h. Then, 15 mM $MgCl_2$ was added, followed by 20-min incubation at room temperature. Excess EDTA and unbound nucleotides were subsequently removed by overnight dialysis against ITC buffer using 10 kDa MWCO Slide-A-Lyzer MINI Dialysis Devices (Thermo Fisher Scientific). Before all measurements, RIPb-CC2, RACB-GDP, RACB-GMPPNP, and RACB-GTPγS were buffer-exchanged by dialysis into the same buffer containing 20 mM HEPES pH 7.5, 100 mM NaCl, 1 mM $MgCl_2$ and 5 mM β-ME. For titrations, RIPb-CC2 was used at a concentration of 20 µM in the cell, while the syringe contained either RACB-GDP, RACB-GMPPNP or RACB-GTPγS at a concentration of 200 µM. Analysis and curve fitting were done with the Malvern PEAQ-ITC software.

### Microscale thermophoresis (MST)

RACB variants were fluorescently labeled with the amino-reactive dye Red-NHS 2nd generation, following the manufacturer's instructions (NanoTemper Technologies, Munich, Germany). A 10 µM solution of RACB (in GDP-, GMPPNP-, GTPγS-bound forms), in labeling buffer (20 mM HEPES pH 7.5, 100 mM NaCl, 5 mM $MgCl_2$, 2 mM TCEP, 0.05% Tween-20, and 10 µM nucleotide), was mixed with a fivefold molar excess of dye. The mixture was incubated for 30 min at room temperature in the dark. Unreacted dye was then removed using a NAP-5 column supplied with the kit. The protein was eluted stepwise with the labeling buffer. Fractions were measured at 280 nm for protein and 650 nm for dye using a nanodrop to determine concentrations. The fraction with the highest protein concentration and degree of labeling was used in binding assays. A series of 24 serial dilutions was prepared, with RIPb-CC2 reaching the highest concentration of 50 µM. The concentration of labeled RACB remained at 10 nM in all experiments. Binding affinities were measured in premium treated capillaries using a Monolith NTX instrument (NanoTemper, Munich, Germany), with 100% infrared laser power and 590 nm LED excitation. The TRIC (Temperature-Related Intensity Change) signals, or the spectral shift (ratio of fluorescence signals at 650 and 670 nm), are plotted against ligand concentration. Data were analyzed with MO.Affinity analysis software v3.0.5 (NanoTemper) to obtain dissociation constants $K_D$. Each binding measurement was performed n = 3 times with separate dilution series. The three datasets were used to obtain average values and error bars. The average dissociation constants $K_D$ and SD were obtained using the three individually analyzed binding experiments for each sample.

### Protein crosslinking

For chemical crosslinking, RACB and RIPb-CC2 in 20 mM HEPES pH 7.5, 100 mM NaCl, 1 mM $MgCl_2$ and 2 mM DTT were combined at a 1:1 molar ratio to reach a final concentration of 20 µM in a total reaction volume of 20 µL. An amino-specific crosslinker, $BS^3$ (Thermo Fisher Scientific), was added at a 1:40 crosslinker-to-protein ratio, and the mixture was incubated for 30 min at room temperature[83]. The reaction was quenched by adding 50 mM Tris-HCl (pH 7.5), followed by a 15 min incubation at room temperature. The samples were analyzed using SDS-PAGE. Negative controls included the protein mixture without $BS^3$, as well as each protein with and without the crosslinker.

### Molecular dynamics (MD) simulations

MD simulations with RACB in complex with GDP or GMPPNP were set up with the solution builder feature of the CHARMM-GUI webserver[84,85] in an orthogonal simulation box at T = 303 K and ambient pressure (1 atm) (see Supplementary Table 1). Long-range Coulomb interactions were accounted for by the particle mesh Ewald method[86]. The NPT ensembles were sampled at an ambient pressure of 1.0 bar, employing the c-rescale isotropic barostat[87] and temperature control was done by v-rescale method[88]. The simulations were conducted for 10 µs with the CUDA-enhanced version of GROMACS (v.2025.2)[89] using the CHARMM36m forcefield[90] on an in-house GPU workstation. Analysis of root mean square fluctuations (r.m.s.f.) values was done with standard GROMACS (v.2025.2) scripts using n = 3 independent simulations (with different seed values) to assess reproducibility and convergence. The construction of a dimeric RACB-RIPb-CC2 structural model at the membrane surface was done in ChimeraX[91] and the membrane builder feature of the CHARMM-GUI webserver[84,85]. We used a 2:2 ratio of RACB-GTP and RIPb-CC2 in POPC:POPG (1:1) lipids and attached a farnesyl moiety at each RACB (via Cys194) monomer using the CHARMM36m forcefield. The complex geometry and lipid surface interaction were then optimized by a 400 ns MD simulation at 303 K using the CUDA-enhanced version of GROMACS (v.2025.2) on an in-house GPU workstation. This run was not used for extracting quantitative information but rather for visualizing the protein complex attached to the lipid membrane surface. Thus, the final structural model was only used for figure preparation using ChimeraX.

### Biological material and growth conditions

Wild-type barley (*Hordeum vulgare L.* cv. Golden Promise) was cultivated in a controlled growth chamber (Conviron, Winnipeg, Canada) at 18 °C with 65% relative humidity under a 16 h light/8 h dark regime (150 μmol m$^{-2}$ s$^{-1}$).

### Cloning procedures for yeast two-hybrid and *in planta* assays

*Hv*RIPb (HORVU1Hr1G012460) was amplified from cDNA introducing *Eco*RI and *Bam*HI restriction sites, respectively[41]. The amplified products were ligated into the pGEM-T easy vector (Promega) by blunt-end cloning according to the manufacturer's instructions and sequenced. Mutagenic primers introducing the double substitution Q540L/W541G (Fw: 5'-GCGCCGGCTGCGCGTGCAGTCCGACCTGGGGCGCAAGGCTGCA GAGGCCGCCG-3'; Rv: 5'-CGGCGGCCTCTGCAGCCTTGCGCCC-CAGGTCGGACTGCACGCGCAGCCGGCGC-3') were used. For yeast two-hybrid (Y2H) assays, pGADT7-RIPb (prey clone) and pGBKT7-CA RACB and pGBKT7-RACB WT (bait clones) were used[41]. The mutant construct pGADT7-RIPbQ540L/W541G was generated by subcloning RIPbQ540L/W541G from the pGEM-T Easy vector (Promega) into the pGADT7 vector (Clontech Laboratories) using *Eco*RI and *Bam*HI restriction sites. For bimolecular fluorescence complementation (BiFC) assays, RIPbQ540L/W541G was PCR-amplified from the pGEM-T Easy vector using primers containing *Spe*I and *Sal*I restriction sites (RIPb-SpeI_Fw: TACTAGTTTCATGCAGAACTC AAAAACCAGTAG and RIPb-Sal-I_Rv: AGTCGACCGGTCTCATGAGCT, respectively). The resulting fragment was digested with *Spe*I and *Sal*I and ligated into the pUC-SPY-NE(R)173 vector[92]. Cloning of pUC-SPYCE-CA RACB, pUC-SPYCE-DN RACB, and pUC-SPYNE-RIPb constructs were carried out with an identical strategy[41].

### Yeast two-hybrid assays

Targeted yeast two-hybrid (Y2H) experiments were carried out using the pGADT7 vector for prey and the pGBKT7 vector for bait, each harboring the respective gene constructs. The plasmids were introduced into the Saccharomyces cerevisiae strain AH109 by the lithium acetate (LiAc)–mediated transformation method, following the manufacturer's recommendations (Clontech Yeast Protocol Handbook).

### Transient transformation of barley epidermal cells

Transient expression in barley (*Hordeum vulgare* cv. Golden Promise) epidermal cells was achieved by particle bombardment. Seven-day-old primary leaves were used for transformation with a PDS-1000/HE particle delivery system (Bio-Rad). For each bombardment, 11 μl of gold particle suspension (1 μm diameter, 27.5 μg/ml; Bio-Rad) was coated with 1 μg of plasmid DNA containing the construct of interest together with 0.5 μg of marker DNA (free mCherry). DNA coating was carried out by adding CaCl$_2$ to a final concentration of 0.5 M and supplementing with 3 μl protamine (2 mg/ml, Sigma), followed by a 30 min incubation at room temperature. The coated particles were then washed sequentially with 70% ethanol and 100% ethanol, resuspended in 6 μl absolute ethanol per shot, and subsequently applied to macrocarriers for bombardment[93].

### Bimolecular fluorescence complementation (BiFC)

BiFC analyses were performed in barley epidermal cells transformed by bombardment with pUC-SPYNE and pUC-SPYCE constructs encoding the N- and C-terminal halves of YFP fused to proteins of interest. A cytosolic mCherry construct was co-expressed in all experiments and served as both a transformation marker and internal reference for normalization. At 24 h post-bombardment, leaf epidermal cells were imaged using confocal laser scanning microscopy under identical acquisition settings. Images were acquired using a Leica TCS SP5 confocal laser scanning microscope. YFP was excited with a 514 nm Argon laser line and detected at 525-570 nm; and mCherry with a 561 nm DPSS diode laser and detected at 570-620 nm. mCherry was recorded using photomultipliers (PMTs), whereas YFP was analyzed with hybrid detectors (HyDs) (Leica Microsystems). Barley epidermal cells were imaged by sequential scanning as z-stacks of single XY optical sections, with 1 μm z-step sizes. Fluorescence quantification was performed as in Fiji[94] by measuring the mean fluorescence intensity (MFI) across entire transformed cells in the YFP and mCherry channels. The BiFC output was expressed as the ratio of YFP to mCherry MFI (MFI_YFP/ MFI_mCherry), thereby correcting for variability in transformation efficiency and protein expression levels. For each construct, at least 20 cells were analyzed per experiment, and the data represent two independent biological replicates.

### Statistics and reproducibility

All statistical parameters were obtained with a minimal data size of n = 3. Values are reported as the mean value ± standard deviation. All in vitro data were measured in at least 3 technical replicates. For the cellular bimolecular fluorescence complementation (BiFC) assays, signals were measured in 10 cells (10 biological replicates) for each construct. Statistical significance was assessed by one-way ANOVA (Tukey's multiple comparison test; $P = 0.05$).

### Reporting summary

Further information on research design is available in the Nature Portfolio Reporting Summary linked to this article.

### Data availability

The atomic coordinates of RACB in different nucleotide-bound forms and in complex with RIPb have been deposited at the Protein Data Bank (PDB) under accession codes 9T3C (RACB-GDP), 9T3D (RACB-GMPPNP), 28NN (RACB-GTPγS), 9T3E (RACB-GTPγS-RIPb-CC2 antiparallel), 9T3F (RACB-GTPγS-RIPb-CC2 parallel). All molecular dynamics simulation data are provided at the zenodo repository (https://doi.org/10.5281/zenodo.18890360). All other raw data used for figure preparation are available as Supplementary Data. Uncropped SDS-PAGE gel images are shown in Supplementary Figs. S16–S18.

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

## Acknowledgements

We acknowledge spectrometer time at the Bavarian NMR Center (BNMRZ) (www.bnmrz.org), Drs. Gerd Gemmecker, Sam Asami, and Matthias Brandl for general NMR support, Florian Rührnößl for HDX-MS measurements, Oktay Göcenler for initial construct cloning, Franziska Scholz and Idil Köker

for protein production. We are grateful for the use of the X-ray Crystallography Platform at Helmholtz Munich. We thank NanoTemper Technologies for providing the material and access to the Monolith instrument, and Dr. Melina Daniilidis and Ivana Jaser for technical support.

## Author contributions
M.M, M.B., R.J., U.G., M.T., C.M, S.M.K., and F.H. conducted research and analyzed data. M.M., M.B., R.J., R.H., and F.H wrote the manuscript. All authors commented on and approved the manuscript. M.M. and F.H. designed the study. D.N., R.H., and F.H acquired funding.

## Funding
This study was supported by the German research foundation (DFG) (project 466160427). The HDX-MS facility was supported by the DFG via the CRC1035 (project Z1, number 201302640), and the TUM Center for Functional Protein Assemblies. ROP work in the laboratory of R.H. was supported by the DFG (HU886/12-1). Open Access funding enabled and organized by Projekt DEAL.

## Competing interests
The authors declare no competing interests.
