## [Transparent Peer Review file · Communications Biology]

Nucleotide-dependent switching and RIPb effector recognition of the barley susceptibility factor RACB

Corresponding Author: Professor Franz Hagn

Version 0:

Reviewer comments:

Reviewer #1

(Remarks to the Author)

Review to the research article "Nucleotide-dependent switching and effector recognition of the barley susceptibility factor RACB"

The manuscript provided by Mohamadi and colleagues describes the structural characterisation of the small GTP-binding protein (GNBP) RACB from barley. RACB belongs to the ROP-family of plant Rho GNBP. It was reported earlier that RACB is activated at the membrane upon fungal infection localizing to the membrane and to microtubules. The authors solved several crystal structures, i.e. RACB in complex with GDP and the PTG-analogue GppNHp and RACB•GTP S in complex with a section of the coiled-coil domain of the effector RIPb, i.e. Ripb-CC2. Conformational changes assessed by NMR and HDX-MS occurring in RACB upon binding to GDP and GppNHp and GTP S suggest that similar as described for proteins of the Ras-family that the switch regions adopt a stable conformation only in the GTP-bound conformation. Interestingly, these investigations also show differences in the conformations of GTP S and GppNHp loaded RACB. The authors complement these studies by assessing the binding affinity of RACB towards Ripb. The interface between RACB and Ripb was confirmed by mutational analyses revealing a sequence motif containing Gln-Trp to be the major specificity determining patch. Studies in plant cells with a split-YFP reporter supports the findings. Overall, the Mohamadi et al. present a well-written manuscript containing very interesting data, which will be highly interesting for the research community working on small GNBP and regulation of infection of plant cells. The manuscript is well-written and the experiments and the conclusions drawn from the results obtained are sound. To this end, I recommend publication of this nice piece of work in Communications Biology. However, I have some minor points that should be addressed in a minor revision prior to publication.

Point 1: Alignment of RACB sequence with other ROP-proteins and human Rho-proteins (RhoA,-B,-C, Cdc42, Rac1). To obtain a better overview on the conservation of sequence patches in RACB compared to other ROP- and Rho-family proteins, it would be helpful to add an alignment in the Supplementary Information. Along that line, the authors should mention the conserved sequence motifs for G1-G5 of RACB in the introduction. Only for the G1/P-loop the authors give the consensus sequence for RACB in the introduction.

Point 2: The authors should also show the amino acid sequences for RACB and Ripb in the Supplementary Information together with the uniprot-accession numbers.

Point 3: In the introduction it is not explained how the infection of the fungus elicits nucleotide exchange of RACB. Is the infection activating a RACB-GEF at the membrane? How is that achieved? Is a transmembrane receptor activated upon fungal infection, which in turn recruits a GEF to the membrane resulting to nucleotide-exchange on RACB at the membrane? The authors could add this information and maybe also include this in Fig. 1a.

Point 4: In Fig. 1a it could be included that GEF exchanges the bound GDP to GTP. The nucleotides are almost not visible.

Point 5: In Fig. 1f the figure legend states that RACB is GppNHp or GTPS loaded, while the legend in the graph says that the RACB is GTP-loaded. Please clarify what is correct and adjust the figure. In the text it is written that there is conformational dynamics in the insert helix in Rop9 and RACB, however, the figure suggests that only in Rop9 there are major

conformational differences in the insert helix in GDP versus GTP-loaded Rop9, while in RACB it is not. Please comment and correct.

Point 6: Throughout the text the authors should write all genus and species names (*Arabidopsis thaliana*, *Arabidopsis*, *Oryza sativa*, etc.) in italics and make it consistent

Point 7: Introduction line 55 describes the CXXL-motif. This motif is known as CaaX-box in proteins of the Ras-family, which is needed for recognition by prenyltransferases. The authors should also use the nomenclature. The C-terminal Leu determines specificity for farnesyltransferase in human Ras-proteins. Is that true also in plants? The proteins are subsequently modified by proteolytic cleavage of the three terminal residues and final carboxymethylation. Is that realized also in plants?

Point 8: The authors say that RACB can be farnesylated or palmitoylated. Farnesylation would be irreversible as thioester, while palmitoylation reversible by thioester. Is it correct that CxxL corresponding to type I is farnesylated at the Cys and GC-CG corresponding to type II is palmitoylated in ROP-proteins? Is it also important that one lipidation is irreversible and the other reversible as also known for human Ras-proteins for processes such as localization to different sub-membranes as described for Ras via a palmitoylation-depalmitoylation cycle? This could be explained a bit clearer in the introduction.

Point 9: Results line 133, typo rho insert region

Point 10: Results lines 137-139, twice "structures" in sentence

Point 11: It is a bit unusual that no electron density for Mg²⁺ was observed in the structures. Is that true for all structures also for RACB•RipB -CC? Mg²⁺ is needed for high-affinity nucleotide binding for both, GDP and GTP. Might that be a result of nucleotide exchange using EDTA? Or does Ripb show any GEF activity towards RACB?

Point 12: The interaction motif is quite small if only the Gln-Trp are essential for the interaction of RACB and Ripb. How is specificity created in that case? Does also the 3D-structure, i.e. presence in the α -helix in the coiled-coil domain important for the interaction or is the sequence sufficient for binding.

Point 13: There is some discrepancy of the data obtained for the affinity between RACB and Ripb by ITC and MST. The ITC data suggests an almost 20-fold weaker affinity (1.7 μ M versus 75 nM). When looking at the ITC data the heat signal obtained, i.e. ΔH , seems to be really large with almost -30 kcal mol⁻¹. That is unusual for a protein-protein interaction. When reading the material and methods section it is obvious that the nucleotide exchange was done by incubating RACB with EDTA and a molar excess of GTP S. It would be good to repeat the ITC experiment with GTP S- or GppNHp-loaded RACB prepared by nucleotide exchange with EDTA and alkaline phosphatase, buffer exchange by size-exclusion chromatography (SEC) or PD10-column to remove the excess nucleotide and EDTA.

Point 14: To analyse which affinity is correct the author could analyse by analytical SEC on a calibrated column if RACB and Ripb co-elute. If yes, probably the nanomolar affinity is correct.

Point 15: Do the authors observe an impact on the oligomerisation of Ripb upon RACB-binding? If the dimer/coiled-coil is resolved that might explain the large ΔH observed. This could also be analysed by analytical SEC.

Point 16: The authors describe that they used constitutively activated RACB or dominant-negative RACB in the cell studies. Please define which mutations were used.

Point 17: In the discussion the authors suggest that differences in the conformation of the insert helix would explain differences in pathways between dicots and monocots. This should be explained a bit more precise to be understandable.

Point 18: In the discussion, the authors compare GppNHp-loaded RACB and GTP S-loaded RACB and suggest that GTP S-loading would stabilize RACB in the fully active conformation. This conclusion cannot be drawn as the structure RACB•GTP S is only available in the complex form with Ripb and the GppNHp-loaded form only in the uncomplexed form. It is not clear if just the binding of Ripb stabilizes RACB in this conformation. To finally conclude this, a complex structure with RACB•GppNHp and/or uncomplexed RACB•GTPS is needed. The authors alter this accordingly.

Point 19: The authors present a model of membrane-bound RACB-Ripb, in which Ripb is in the parallel architecture. How about the anti-parallel orientation of the helices in the coiled-coil domain. Is that less likely?

Point 20: In Fig. S1 the axes of figures showing the SEC runs are not labelled. In SEC runs "a.u." means „absorbance units" and should be added.

Point 21: The SEC runs in Fig S1 (b,c) are not performed on a suitable column. I assume a S75 column was used. Please add this information in the figure legend. At 40 mL elution volume a void volume peak is visible in both figures, i.e. S1b and S1c. That suggests that the Ripb protein is to some extent aggregated or forming a higher oligomer. Was the column calibrated? Can the authors suggest if RCB forms a monomer and Ripb a dimer with a sub-fraction being aggregated/forming a higher oligomer. With a S200 column the authors could separate the void volume peak from the proper dimer-peak.

Point 22: Please show the N-values, H, T S and KD-values for the ITC data.

Reviewer #2

(Remarks to the Author)

The paper by Hagn and coworkers presents a comprehensive report on the plant small GTPase RACB, its structure in different activity states, its dynamics and binding to an effector RIPB. This GTPase had no been extensively studied and the work is of significance due to the comparisons which are made with homologous GTPases in plants as well as with human Rho GTPases.

The work is overall well executed and the paper is well written. Remarkable is the wide variety of techniques which have been employed (ranging from CD to NMR, crystallography to HDX-MS and cell based studies) and the level of completion of the project. I only have relatively minor requests for clarification and additions.

GMPPNP is not used for some of the experiments, e.g. cross-linking, MST and ITC. Why is that?

For the CD thermal melts please give time for heat up, or rather time of collection of each data points+time between points as the unfolding of GTPases is typically not-reversible. Comment on the stability of GMPPNP in the protein..under some conditions esp. pH and temperature this does hydrolyze.

Since relaxation measurements are very sensitive to even non-specific intermolecular interactions, what was done to ascertain that RACB is monomeric throughout?

For the simulations, how many replicas were run? If only one they should mention this as a caveat.

Please describe the exact C-terminal lipidation which was done and the forcefield which was available for this or had to be parameterized.

Fig. 1f value of -2, a better designation of sequence gaps should be given, like a *, not linking the data.

In SI please given sequence alignment for RACB, Rop9 and Rac1 (maybe also the closest human rho gtpase).

Line 251 needs attention, RACB-GDP has lower HetNOE values at the C- but not N-terminus.

A comparison between simulation RMSF and B-factors may be informative as there is a formula that converts between them. E.g. see <https://pubmed.ncbi.nlm.nih.gov/16361340/>

Line 269 GMPPNP-RACB vs. RACB-GMPPNP; use latter for consistency across MS.

Line 299 and 300 "with" rather than "to"

Line 341, Fig. 4 mention grey shading in b) and d)

Line 397 and 400 – the mutants for CA and DN should be given

Line 421-424 add a few details about the modeling, simulations

Fig. 6 d) mention yellow at RIPb C-term. in e) it is unclear what the spheres are (lipids?) and why the C-term is now in the membrane/near the outer surface f) mention that this is a view top down from the membrane.

Lines 525-529 explain more how that study targeted only one signal pathway.

SI section:

Nucleotide exchange: mention RACB concentration during the exchange. Later- not clear what the respective buffer is.

HDX-MS: a figure is needed to show sequence coverage and also overlap between peptides

MD simulations: please state the version of charm-gui and the forcefield used.

Fig. S4 last line of legend "The assignments in the GMPPNP form of resonances missing in the GTPgammaS bound state are labeled."

Fig. S6 provide other data from ITC fits, e.g. n, deltaH

Fig. S8 why are the two y-axes different?

Reviewer #3

(Remarks to the Author)

Note to Authors: The study conducted by Mohamadi et al., focuses on the characterization of barley ROP GTPase RACB using different structural techniques like X-ray crystallography, NMR, and HDX mass spectrometry. The article provides key information regarding the nature of the interaction of nucleotide-bound forms of RACB with its binding partner RIPb. Overall, the article is clearly written and can be easily understood. However, there are a few areas where the study can be further improved.

Comments:

Q1. What is the motivation behind using both GMPPNP and GTPgammaS for thermal stability, CD, and NMR studies, when the crystal structure was solved with GMPPNP? There seems to be no explanation why the authors chose to solve the crystal structure of active RACB only with GMPPNP.

Q2. The reported overall B-factor seems to be high for the GDP-bound state as compared to the GMPPNP-bound state (Table S1), whereas the thermal stability curves of RACB with GDP and GMPPNP are very similar (Figure S2b). Can the authors account for a reason?

Q3. Line 139: The sentence ends with ‘.’. This should be rectified.

Q4: Fig 1f: Apart from providing a C-alpha displacement plot, a sequence alignment highlighting the regions where there are significant differences will be a helpful addition.

Q5: Line 157: To understand the conformational differences between the two structures, Fig 1d is the appropriate one. The authors have mentioned Fig 1c, f in parentheses, although neither Fig 1c nor Fig 1f highlights the conformational differences between the two structures.

Q6: Did the authors want to compare the GDP-RACB with GDP-ROP9 and GMPPNP-RACB with GMPPNP-RAC1? If so, then the legend of Figure 1 should be revisited. The explanation (Lines 171-183) does not correlate with the legend section of Figure 1.

Q7. Line 468-469: The authors claim that the high-resolution structures of RACB in the different nucleotide-bound forms shed light on activation from inactive – pre-active - fully active forms. It can be understood that GDP bound form is the inactive form, but it remains elusive on what evidence the authors have differentiated between GMPPNP and GTPgammaS as analogs for pre-active form and the fully active form. This needs more explanation.

Q8. In Figure S8, the two graphs have been plotted with two different Y-axes. If the graphs are to be compared, then both should be plotted on the same axes. Secondly, if a comparison has to be drawn between the affinity of GTPgammaS and GMPPNP-bound RACB to bind RIPb, then other complementary techniques should be used besides MST. If RACB-GTPgammaS binds RIPb better than RACB-GMPPNP, then evaluation by other techniques is required to draw firm conclusions.

Q9. Although computational methods have been employed to show that RACB-RIPb interactions are associated with membrane interactions, experimental evidence is required to support the hypothesis. Binding experiments of nucleotide-bound RACB with plant lipids in the presence and absence of RIPb will further the understanding of the structural basis to functional significance.

Version 1:

Reviewer comments:

Reviewer #1

(Remarks to the Author)

Review of the revised manuscript entitled “Nucleotide-dependent switching and effector recognition of the barley susceptibility factor RACB” submitted by Mohamadi and colleagues.

Overall, the current manuscript is a really comprehensive analyses on the small GTP-binding protein RacB from barley. The authors present a sound study and it further improved during this round of revision. The authors performed additional experiments, including analytical size-exclusion chromatography, CD spectroscopy and they even solved another crystal structure of RacB•GTPgS to resolve the few open points. The authors have completely addressed all open points and I highly recommend publication of this manuscript.

Reviewer #2

(Remarks to the Author)

The manuscript has been reviewed in considerable depth by three reviewers and I find that my comments and it appears on quick reading also those of the other reviewers have been adequately addressed.

One minor comment: when gel-filtration is mentioned as evidence for non-aggregation please estimate and state the protein concentration on the column by absorbance.

Reviewer #3

(Remarks to the Author)

Thanks to all three reviewers for their positive assessment of our work and the constructive comments that helped us to further improve our manuscript. With all the changes to the manuscript and the inclusion of a large set of additional data, we hope that all remaining questions are fully addressed.

To be able to quickly check the changes made in the manuscript, we refer the reviewers to the uploaded marked version with tracked changes. We anticipate that this procedure is easier than including all revised pieces of text and figures in the response letter here.

Reviewer #1 (Remarks to the Author):

Review to the research article “Nucleotide-dependent switching and effector recognition of the barley susceptibility factor RACB”

The manuscript provided by Mohamadi and colleagues describes the structural characterisation of the small GTP-binding protein (GNBP) RACB from barley. RACB belongs to the ROP-family of plant Rho GNBP. It was reported earlier that RACB is activated at the membrane upon fungal infection localizing to the membrane and to microtubules. The authors solved several crystal structures, i.e. RACB in complex with GDP and the PTG-analogue GppNHp and RACB•GTP□S in complex with a section of the coiled-coil domain of the effector RIPb, i.e. Ripb-CC2. Conformational changes assessed by NMR and HDX-MS occurring in RACB upon binding to GDP and GppNHp and GTP□S suggest that similar as described for proteins of the Ras-family that the switch regions adopt a stable conformation only in the GTP-bound conformation. Interestingly, these investigations also show differences in the conformations of GTP□S and GppNHp loaded RACB. The authors complement these studies by assessing the binding affinity of RACB towards Ripb. The interface between RACB and Ripb was confirmed by mutational analyses revealing a sequence motif containing Gln-Trp to be the major specificity determining patch. Studies in plant cells with a split-YFP reporter supports the findings. Overall, the Mohamadi et al. present a well-written manuscript containing very interesting data, which will be highly interesting for the research community working on small GNBP and regulation of infection of plant cells. The manuscript is well-written and the experiments and the conclusions drawn from the results obtained are sound. To this end, I recommend publication of this nice piece of work in Communications Biology. However, I have some minor points that should be addressed in a minor revision prior to publication.

Thanks to this reviewer for their very positive assessment of our work.

Point 1: Alignment of RACB sequence with other ROP-proteins and human Rho-proteins (RhoA,-B,-C, Cdc42, Rac1). To obtain a better overview on the conservation of sequence patches in RACB compared to other ROP- and Rho-family proteins, it would be helpful to add an alignment in the Supplementary Information. Along that line, the authors should mention the conserved sequence motifs for G1-G5 of RACB in the introduction. Only for the G1/P-loop the authors give the consensus sequence for RACB in the introduction.

We included a short paragraph into the introduction on the sequence motifs and added a sequence alignment figure of the suggested small GTPase proteins into the Supporting Information as Figure S1.

Point 2: The authors should also show the amino acid sequences for RACB and Ripb in the Supplementary Information together with the uniprot-accession numbers.

This information is now part of the Supplementary Methods section (first paragraph).

Point 3: In the introduction it is not explained how the infection of the fungus elicits nucleotide exchange of RACB. Is the infection activating a RACB-GEF at the membrane? How is that

achieved? Is a transmembrane receptor activated upon fungal infection, which in turn recruits a GEF to the membrane resulting to nucleotide-exchange on RACB at the membrane? The authors could add this information and maybe also include this in Fig. 1a.

Thanks. We added further information to this part of the introduction. Even though it is not entirely clear at this point, the activation of RACB could happen via the plant-specific ROP-GEF14. We also included this scenario in Fig. 1a.

Point 4: In Fig. 1a it could be included that GEF exchanges the bound GDP to GTP. The nucleotides are almost not visible.

We now labeled the nucleotide-states in Fig. 1a in a clearer manner and explained GEF (see comment above) and GAP in the figure legend. With this, we believe that the action of a GEF and GAP can be understood without further explanations.

Point 5: In Fig. 1f the figure legend states that RACB is GppNHp or GTPS loaded, while the legend in the graph says that the RACB is GTP-loaded. Please clarify what is correct and adjust the figure. In the text it is written that there is conformational dynamics in the insert helix in Rop9 and RACB, however, the figure suggests that only in Rop9 there are major conformational differences in the insert helix in GDP versus GTP-loaded Rop9, while in RACB it is not. Please comment and correct.

Thanks for this comment. We updated figure 1f accordingly to be consistent with the correct figure legend. At this point we do not discuss conformational dynamics, but rather conformational changes as seen in the crystal structures used for the comparison. We explicitly state in the text that the insert region in RACB does not change its structure but there are changes when comparing Rop9 and Rac1.

Point 6: Throughout the text the authors should write all genus and species names (*Arabidopsis thaliana*, *Arabidopsis*, *Oryza sativa*, etc.) in italics and make it consistent

Thanks. Has been fixed

Point 7: Introduction line 55 describes the CXXL-motif. This motif is known as CaaX-box in proteins of the Ras-family, which is needed for recognition by prenyltransferases. The authors should also use the nomenclature. The C-terminal Leu determines specificity for farnesyltransferase in human Ras-proteins. Is that true also in plants? The proteins are subsequently modified by proteolytic cleavage of the three terminal residues and final carboxymethylation. Is that realized also in plants?

We updated the corresponding section. Concerning the mechanistic question: This is conserved in plants. See: Yalovsky, S. (2015). Protein lipid modifications and the regulation of ROP GTPase function. *Journal of experimental botany*, 66(6), 1617-1624, which is cited in the manuscript (Ref 20).

Point 8: The authors say that RACB can be farnesylated or palmitoylated. Farnesylation would be irreversible as thioester, while palmitoylation reversible by thioester. Is it correct that CxxL corresponding to type I is farnesylated at the Cys and GC-CG corresponding to type II is palmitoylated in ROP-proteins?

This general statement on type I and type II RAC/ROP proteins does not apply entirely to RACB. RACB is type I and is thus irreversibly farnesylated only. See Ref 20 and the updated text in the introduction. RACB membrane association and CAAX BOX importance is shown in: Weiß et al, *Plant J*, 2025, also cited in the manuscript.

Is it also important that one lipidation is irreversible and the other reversible as also known for human Ras-proteins for processes such as localization to different sub-membranes as described for Ras via a palmitoylation-depalmitoylation cycle? This could be explained a bit clearer in the introduction.

Thanks for this comment. RACB has only one prenylation site via a cysteine thioether linkage and no second lipidation site. The importance of the CAAX Box and RACB membrane association has been shown in a recent paper (<https://doi.org/10.1111/tpj.70356>) and is also described in Ref 20. In addition to farnesylation, electrostatic interactions via the polybasic stretch with negatively charged phosphatidylinositol-monophosphates are essential for stable membrane interaction. We added a specific statement on RACB membrane interaction with the above-mentioned reference. With this, we hope that this addition facilitates clarity.

Point 9: Results line 133, typo rho insert region

Has been fixed.

Point 10: Results lines 137-139, twice “structures” in sentence

Has been fixed.

Point 11: It is a bit unusual that no electron density for Mg²⁺ was observed in the structures. Is that true for all structures also for RACB•RipB-CC? Mg²⁺ is needed for high-affinity nucleotide binding for both, GDP and GTP. Might that be a result of nucleotide exchange using EDTA? Or does Ripb show any GEF activity towards RACB?

We do observe a very clear electron density corresponding to Mg²⁺ bound at its canonical position. The magnesium ion was included during refinement and is present in the structures deposited in the PDB for RACB–GDP (PDB ID: 9T3C), RACB–GTPγS (PDB ID: 28NN), RACB–GTPγS–RIPbCC2 (parallel orientation; PDB ID: 9T3F), and RACB–GTPγS–RIPbCC2 (antiparallel orientation; PDB ID: 9T3E). The only exception is the RACB–GMPPNP complex (PDB ID: 9T3D), in which the Mg²⁺ binding site is occupied by the side chain of Lys182 from a symmetry-related molecule, representing a crystal-packing artefact. In all samples, excess Mg²⁺ was present and we are not aware of a potential GEF activity of RIPb.

Point 12: The interaction motif is quite small if only the Gln-Trp are essential for the interaction of RACB and Ripb. How is specificity created in that case? Does also the 3D-structure, i.e. presence in the α-helix in the coiled-coil domain important for the interaction or is the sequence sufficient for binding.

In both RACB–GTPγS–RIPbCC2 structures - parallel (PDB ID: 9T3F) and antiparallel (PDB ID: 9T3E) - we observe an extensive interaction network between the two proteins. The parallel complex is stabilized by approximately ten hydrogen bonds and salt bridges, together with numerous hydrophobic contacts, resulting in a total buried interface area of ~530 Å². Similarly, the antiparallel complex contains ~12 hydrogen bonds and salt bridges, complemented by hydrophobic interactions, with a buried surface area of ~500 Å². These observations indicate that, once formed, the RACB–RIPbCC2 complex is stabilized by a substantial and well-defined protein–protein interface.

Despite this extensive interface, microscale thermophoresis (MST) experiments using the Q540L/W541G RIPb-CC2 variant failed to detect binding to RACB, demonstrating that disruption of the Gln-Trp motif abolishes the interaction. Together with previous observations in *Arabidopsis* (Lavy et al., 2007), this strongly suggests that the Gln-Trp tandem constitutes the primary recognition and specificity determinant for RACB

binding. We therefore propose that specificity is initially established through this short sequence motif, while the surrounding coiled-coil α -helical structure contributes to the correct spatial presentation and stabilization of the interaction interface once binding has been initiated.

Point 13: There is some discrepancy of the data obtained for the affinity between RACB and Ripb by ITC and MST. The ITC data suggests an almost 20-fold weaker affinity (1.7 μ M versus 75 nM). When looking at the ITC data the heat signal obtained, i.e. ΔH , seems to be really large with almost -30 kcal mol⁻¹. That is unusual for a protein-protein interaction. When reading the material and methods section it is obvious that the nucleotide exchange was done by incubating RACB with EDTA and a molar excess of GTP γ S. It would be good to repeat the ITC experiment with GTP γ S- or GppNHp-loaded RACB prepared by nucleotide exchange with EDTA and alkaline phosphatase, buffer exchange by size-exclusion chromatography (SEC) or PD10-column to remove the excess nucleotide and EDTA.

For the ITC sample preparation, we followed established protocols and did use a PD10 column to remove excess nucleotides and EDTA. The exact procedure is now described in more detail in the revised SI method section. The samples for ITC and MST were prepared in the same way. The systematic difference in the obtained K_D values is thus very likely originating from the hydrophobic fluorescence dye on RACB that is required for the MST experiments. We also performed ITC experiments with RACB-GMPPNP and RIPb-CC2 and detected a \sim 15 μ M K_D value (see Fig. S11), which is \sim 8-fold weaker as observed with GTP γ S. This difference follows a similar tendency as seen by MST (75 versus 340 nM).

Point 14: To analyse which affinity is corrects the author could analyse by analytical SEC on a calibrated column if RACB and Ripb co-elute. If yes, probably the nanomolar affinity is correct.

Thanks for this suggestion. We conducted analytical S200 size exclusion chromatography and could only detect a very small portion of RACB coeluting with RIPb-CC2 (Fig. S12). Since this assay is done with non-modified RACB (i.e. no fluorescence dye attached), we believe that the μ M affinity derived from ITC is correct. This is also now stated in the respective results section (p.12).

Point 15: Do the authors observe an impact on the oligomerisation of Ripb upon RACB-binding? If the dimer/coiled-coil is resolved that might explain the large ΔH observed. This could also be analysed by analytical SEC

We addressed this interesting question by measuring CD spectra of RIPb-CC2 at different temperatures (5 versus 20°C, Fig. S10a) and indeed could observe a very pronounced increase in α -helical secondary structure at lower temperature. This suggests that at 20°C RIPb-CC2 has to undergo a folding transition upon binding to RACB giving rise to a larger enthalpy. To further strengthen this hypothesis, we also conducted ITC experiments at different temperatures and could indeed observe that at lower temperature (15°C) the binding enthalpy is very much reduced as compared to the data obtained at 25°C. At the same time the number of binding sites is also almost 1:1 at lower temperature and very low (0.3) at 25°C. We now added these data in Fig. S10b,c and the results section (p.12).

Point 16: The authors describe that they used constitutively activated RACB or dominant-negative RACB in the cell studies. Please define which mutations were used.

CA stands for constitutively active (RACB-G15V) and DN stands for dominant negative (RACB-T20N). This is now specified in the corresponding results section.

Point 17: In the discussion the authors suggest that differences in the conformation of the insert helix would explain differences in pathways between dicots and monocots. This should be explained a bit more precise to be understandable.

We agree with this reviewer that that statement is a bit too vague, and beyond the main message of the present manuscript. Thus, since space in the manuscript is limited and to streamline the writeup, we decided to delete that sentence entirely.

Point 18: In the discussion, the authors compare GppNHp-loaded RACB and GTP γ S-loaded RACB and suggest that GTP γ S-loading would stabilize RACB in the fully active conformation. This conclusion cannot be drawn as the structure RACB•GTP γ S is only available in the complex form with Ripb and the GpNHp-loaded form only in the uncomplexed form. It is not clear if just the binding of Ripb stabilizes RACB in this conformation. To finally conclude this, a complex structure with RACB•GppNp and/or uncomplexed RacB•GTPS is needed. The authors alter this accordingly.

We thank the reviewer for this important and well-taken point. We agree that, based solely on the currently deposited structures, a definitive conclusion that GTP γ S stabilizes RACB in a fully active conformation cannot be drawn, as RACB•GTP γ S is available only in complex with RIPb, whereas RACB•GppNHp was determined in the uncomplexed state.

Thus, we have additionally obtained and analyzed X-ray diffraction data for uncomplexed RACB•GTP γ S at ~3.0 Å resolution. The somewhat lower resolution in the GTP γ S-bound state is consistent with our NMR experiments of GTP γ S-bound RACB (Fig. S7), which indicates increased overall exchange dynamics.

Structural overlays that are now also part of the revised manuscript reveal that the switch regions of uncomplexed RACB•GTP γ S adopt conformations highly similar to those observed in the RACB•GTP γ S-RIPb-CC2 complex. This observation is consistent with the statement that GTP γ S binding favors a more active-like conformation of RACB.

To depict this important conclusion, we here provide a structural comparison illustrating the structural similarity of RACB•GTP γ S and RACB•GTP γ S–RIPbCC2 for the reviewer's assessment. Furthermore, we deposited the new structure in the pdb (pdb:28NN, also uploaded as Supplementary Data for review) and present it in the results section in Figs. 1 and 5.

Point 19: The authors present a model of membrane-bound RACB-Ripb, in which Ripb is in the parallel architecture. How about the anti-parallel orientate of the helices in the coiled-coil domain. Is that less likely?

Considering the geometry for membrane binding, the antiparallel orientation would not bring the membrane binding sites of RACB (incl the prenylation site) to close proximity. Furthermore, the basic C-terminal tail of one Ripb monomer would also point away from the membrane. Thus, even though the coiled coil dimer seems to have some degree of freedom, the only reasonable arrangement is the parallel orientation. Due to these considerations, we used that orientation to model the RACB-RIPb-CC2 complex at the membrane surface.

Point 20: In Fig. S1 the axes of figures showing the SEC runs are not labelled. In SEC runs “a.u.” means „absorbance units” and should be added.

Thanks. This is now fixed in the new Fig. S2.

Point 21: The SEC runs in Fig S1 (b,c) are not performed on a suitable column. I assume a S75 column was used. Please add this information in the figure legend. At 40 mL elution volume a void volume peak is visible in both figures, i.e. S1b and S1c. That suggests that the Ripb protein is to some extent aggregated or forming a higher oligomer. Was the column calibrated? Can the authors suggest if RCB forms a monomer and Ripb a dimer with a sub-fraction being aggregated/forming a higher oligomer. With a S200 column the authors could separate the void volume peak from the proper dimer-peak.

Thanks for this comment. The rationale behind using a S75 column is to be able to directly compare the elution volumes of RACB and RIPb-CC2. The column is calibrated with the Cytiva standards (Ribonuclease A (~13.7 kDa), Chymotrypsinogen A (~25 kDa), Ovalbumin (~43 kDa), Conalbumin (Ovotransferrin) (~75 kDa), where both investigated proteins (~20 and 25kDa, respectively) are well in range. To address this point of concern, we re-ran the SEC75 experiments with more carefully purified samples (as done for all experiments in this study) and obtained better data with elution volumes (away from the void volume at V ~ 45mL) that allowed the calculation of realistic

apparent molecular weights of ~25 kDa for the RACB monomer and ~88 kDa for (elongated) the RIPb-CC2 dimer. Furthermore, we ran the samples also on a S200a (24mL) column (Fig. S12a,b) and obtained similar MW values of 19.7 and 98 kDa, respectively. Together with the chemical crosslinking data (Fig. S12c), it is safe to say that RIPb-CC2 is a dimer in solution.

Point 22: Please show the N-values, DH, TDS and KD-values for the ITC data.

We now report the requested ITC data for RACB-GTP γ S at 20°C derived from three technical replicates in Fig. S9. Also, we added the same set of ITC data for RACB-GMPPNP (Fig. S11a).

Reviewer #2 (Remarks to the Author):

The paper by Hagn and coworkers presents a comprehensive report on the plant small GTPase RACB, its structure in different activity states, its dynamics and binding to an effector RIPB. This GTPase had not been extensively studied and the work is of significance due to the comparisons which are made with homologous GTPases in plants as well as with human Rho GTPases.

The work is overall well executed and the paper is well written. Remarkable is the wide variety of techniques which have been employed (ranging from CD to NMR, crystallography to HDX-MS and cell based studies) and the level of completion of the project. I only have relatively minor requests for clarification and additions.

Thanks to this reviewer for their very positive assessment of our work. We appreciate the constructive comments.

1. GMPPNP is not used for some of the experiments, e.g. cross-linking, MST and ITC. Why is that?

We have now updated the dataset and used GMPPNP for ITC and crosslinking experiments. However, we already had used GMPPNP for MST experiments in the initial manuscript. The rationale behind rather using GTP γ S is simply that it induces a more active conformation in RACB and thus leads to a higher affinity for RIPb-CC2. This statement is nicely corroborated by the new crystal structure of RACB-GTP γ S we now included in the manuscript, where RACB adopts a very similar conformation as in the complex with RIPb-CC2.

2. For the CD thermal melts please give time for heat up, or rather time of collection of each data points+time between points as the unfolding of GTPases is typically not-reversible. Comment on the stability of GMPPNP in the protein..under some conditions esp. pH and temperature this does hydrolyze.

Thanks. This info is now provided in the method section. We also see a non-reversible thermal melting behavior, as is the case for most proteins. We do not see changes in the sample signatures of GMPPNP-bound RACB over time in the NMR spectra and we clearly see the full nucleotide in the crystal structure that has been crystallized for > 1 week at 4°C or 20°C, respectively.

3. Since relaxation measurements are very sensitive to even non-specific intermolecular interactions, what was done to ascertain that RACB is monomeric throughout?

Our SEC data with RACB clearly shows a homogenous peak at an elution volume of ~68 ml on an S75 column (total volume 124 mL). This corresponds to roughly 20 kDa and

the monomeric form. The calculated MW of RACB is now explicitly stated in Fig S2 and Fig S12.

Furthermore, we recorded preliminary ¹⁵N T1 and T2 experiments with RACB-GDP and RACB-GMPPNP. The rigid region between residues 80 and 100 (see hetNOE plot in Fig. S8) was used to calculate average T1 and T2 values for each sample. With the T1/T2 ratio at hand, we estimated rotational correlation times of 7.2 and 6.9 ns for the GDP or GMPPNP-bound forms, respectively. This corresponds to a globular protein of ~19 kDa, which is in almost perfect agreement with the molecular weight of RACB of 19.6 kDa (RACB residues 1-179). We now added a statement to the results section to address this point.

4. For the simulations, how many replicas were run? If only one they should mention this as a caveat.

Thanks for this comment. We have now run 3 individual MD simulations (10 μs each) that nicely reproduce the data shown in Figure S8. Figure S8 has been updated with a mean rmsf value + SD plotted derived from the three independent simulations, as well as a rmsd plot to show that the simulations were running stably.

5. Please describe the exact C-terminal lipidation which was done and the forcefield which was available for this or had to be parameterized.

We used the CHARMM36m forcefield for attaching a farnesyl moiety (available in the FF). This is now also updated in the Supporting Methods.

6. Fig. 1f value of -2, a better designation of sequence gaps should be given, like a *, not linking the data.

Is fixed.

7. In SI please given sequence alignment for RACB, Rop9 and Rac1 (maybe also the closest human rho gtpase).

Thanks for this comment. We have now inserted a new Figure S1 with an MSA.

8. Line 251 needs attention, RACB-GDP has lower HetNOE values at the C- but not N-terminus.

Thanks. We fixed this inaccuracy.

9. A comparison between simulation RMSF and B-factors may be informative as there is a formula that converts between them. E.g. see <https://pubmed.ncbi.nlm.nih.gov/16361340/>

Thanks for this suggestion. We calculated the rmsf values from the X-ray B-factors and could see a quite good correlation for RACB-GMPPNP (see figure below). However, since the overall B-factors were generally higher for RACB-GDP (for reasons outlined below in our answers to the comments by reviewer #3), the calculated rmsf pattern did not agree very well with the MD as well as the NMR results. Thus, we decided not to include such a comparison in the manuscript.

10. Line 269 GMPPNP-RACB vs. RACB-GMPPNP; use latter for consistency across MS.

Done

11. Line 299 and 300 “with” rather than “to”

Thanks. Has been fixed.

12. Line 341, Fig. 4 mention grey shading in b) and d)

Thanks. Has been updated in legend to Fig. 4

13. Line 397 and 400 – the mutants for CA and DN should be given

CA stands for constitutively active (RACB-G15V) and DN stands for dominant negative (RACB-T20N). This is now updated in the manuscript.

14. Line 421-424 add a few details about the modeling, simulations

We now added further information at this point but also refer the reader to the supplementary information methods for more details.

15. Fig. 6 d) mention yellow at RIPb C-term. in e) it is unclear what the spheres are (lipids?) and why the C-term is now in the membrane/near the outer surface f) mention that this is a view top down from the membrane.

Thanks for these comments. We updated the figure legend accordingly to better explain the features of the structural model.

16. Lines 525-529 explain more how that study targeted only one signal pathway.

Thanks for this comment. Since we cannot comment on or predict specific targeting of fungal infection by inhibitors of the RACB-RIPb interaction, we decided to remove the last paragraph.

17. SI section: Nucleotide exchange: mention RACB concentration during the exchange. Later- not clear what the respective buffer is.

Thanks, these details are now mentioned in the SI method section.

18. HDX-MS: a figure is needed to show sequence coverage and also overlap between peptides

This is now included for RACB alone (Fig. S6) and the RACB-RIPb-CC2 complex (Fig. S14)

19. MD simulations: please state the version of charm-gui and the forcefield used.

We used the charm-gui webserver as cited in the manuscript. For the MD run we used the CUDA version of Gromacs 2025.2 (see SI methods) and the CHARMM36m forcefield (now also mentioned in the methods section and Table S1).

20. Fig. S4 last line of legend “The assignments in the GMPPNP form of resonances missing in the GTP γ S bound state are labeled.”

Thanks. Has been fixed.

21. Fig. S6 provide other data from ITC fits, e.g. n, ΔH

Thanks. The numbers are now provided in the legend to Figure S9 and in Figs. S10&S11.

22. Fig. S8 why are the two y-axes different?

In MST there are two modes of readouts, the classical MST with a fluorescence signal detected at one wavelength (650nm) and the (more recent) expansion of using the fluorescence ratio 670/650 nm (spectral shift). Depending on the exact binding mode and affinity, it is a matter of exploration which detection mode is best suited. In our samples, it turned out that the RACB-GMPPNP worked better with the classical detection mode and the GTP γ S complex better with the spectral shift. We now added a statement to clarify this confusion to the legend of Fig. S11. See also the newly cited reference (Ref 34) in Fig. S11 legend. Also, the plot shown below indicates the spectral shift data for the RACB-GMPPNP (dark red) and RACB- GTP γ S (green) interaction with RIPb-CC2 where it can be clearly seen that the data quality for the GMPPNP sample is responding to RIPb-CC2 but cannot be analyzed by a binding model.

Reviewer #3 (Remarks to the Author):

Note to Authors: The study conducted by Mohamadi et al., focuses on the characterization of barley ROP GTPase RACB using different structural techniques like X-ray crystallography, NMR, and HDX mass spectrometry. The article provides key information regarding the nature of the interaction of nucleotide-bound forms of RACB with its binding partner RIPb. Overall, the article is clearly written and can be easily understood. However, there are a few areas where the study can be further improved.

Comments:

Q1. What is the motivation behind using both GMPPNP and GTP γ S for thermal stability, CD, and NMR studies, when the crystal structure was solved with GMPPNP? There seems to be no explanation why the authors chose to solve the crystal structure of active RACB only with GMPPNP.

We now also provide crystal structures of RACB-GTP γ S to have a complete structural picture of all states under investigation. The initial lack of that structure was due to lower crystal quality, presumably caused by enhanced dynamics in RACB if bound to GTP γ S (see Fig. S7). However, we finally managed to refine the structure to 3 Å resolution to enable a solid comparison with the other states, which is now shown in Fig.1. Nonetheless, the thermal stabilities and other biophysical methods nicely show that GTP γ S seems to better fit into the structure leading to a higher stability. Also, the NMR data also show that GTP γ S induced additional line broadening effects as compared to GMPPNP. Thus, we consider the work with GMPPNP valuable and a prerequisite to map GTP γ S-broadened resonances to the structure of RACB (Fig. S7).

Q2. The reported overall B-factor seems to be high for the GDP-bound state as compared to the GMPPNP-bound state (Table S1), whereas the thermal stability curves of RACB with GDP and GMPPNP are very similar (Figure S2b). Can the authors account for a reason?

The two parameters referred to by this reviewer report on fundamentally different physical properties and therefore are not directly comparable.

The overall B-factor reported in Table 1 (average temperature factor) is derived from the X-ray crystal structure and reflects the degree of atomic displacement and structural order within the crystal lattice. It is influenced by factors such as crystal packing, local flexibility, resolution, and refinement, and thus reports on structural order in the crystalline state, not on protein stability or ligand binding affinity. In contrast, the thermal stability curves shown in Figure S3b measure the global thermodynamic stability of RACB in solution, as reflected by the melting temperature. The very similar melting curves observed for RACB bound to GDP and GMPPNP indicate that the overall fold and thermal stability of the protein are essentially identical in solution, regardless of the bound nucleotide.

Q3. Line 139: The sentence ends with ‘..’ .This should be rectified.

Thanks. Has been fixed.

Q4: Fig1f: Apart from providing a C-alpha displacement plot, a sequence alignment highlighting the regions where there are significant differences will be a helpful addition.

A corresponding MSA is now included in the SI part (Fig. S1).

Q5: Line 157: To understand the conformational differences between the two structures, Fig 1d is the appropriate one. The authors have mentioned Fig 1c, f in parentheses, although neither Fig 1c nor Fig 1f highlights the conformational differences between the two structures.

We have now re-arranged Fig. 1 to also include RACB-GTP γ S and updated the figure legend to clearly state what structures are being compared. Also, the structural difference between the other plant GTPase structures is now moved to Fig. S4.

Q6: Did the authors want to compare the GDP-RACB with GDP-ROP9 and GMPPNP-RACB with GMPPNP-RAC1? If so, then the legend of Figure 1 should be revisited. The explanation (Lines 171-183) does not correlate with the legend section of Figure 1.

See comment above. Fig. 1 is now revised, and the overall clarity is improved.

Q7. Line 468-469: The authors claim that the high-resolution structures of RACB in the different nucleotide-bound forms shed light on activation from inactive – pre-active - fully active forms. It can be understood that GDP bound form is the inactive form, but it remains elusive on what evidence the authors have differentiated between GMPPNP and GTP γ S as analogs for pre-active form and the fully active form. This needs more explanation.

As mentioned above, we now not only provide the structure of RACB-GTP γ S in complex with RIPb but also of RACB-GTP γ S alone, which fully confirms the previous statement in the manuscript that the three nucleotides lead to differences in activity monitored also by the increase in binding affinity to the effector RIPb.

Q8. In Figure S8, the two graphs have been plotted with two different Y-axes. If the graphs are to be compared, then both should be plotted on the same axes. Secondly, if a comparison has to be drawn between the affinity of GTP γ S and GMPPNP-bound RACB to bind RIPb, then other complementary techniques should be used besides MST. If RACB-GTP γ S binds RIPb better than RACB-GMPPNP, then evaluation by other techniques is required to draw firm conclusions.

In MST there are two modes of readouts, the classical MST with fluorescence detected at one wavelength (650nm) and the (more recent) expansion of using the fluorescence ratio 670/650 nm (spectral shift). Depending on the exact binding mode and affinity, it is a matter of exploration which detection mode is best suited. In our samples, it turned out that the RACB-GMPPNP worked better with the classical detection mode and the GTP γ S complex better with the fluorescence ratio. We now added a statement to clarify this confusion to the legend of Fig. S11. See also the newly cited reference (Ref 34) in Fig. S11 legend. Also, the plot shown below indicates the spectral shift data for the RACB-GMPPNP (dark red) and RACB-GTP γ S (green) interaction with RIPb-CC2 where it can be clearly seen that the data quality for the GMPPNP sample is responding to RIPb-CC2 but cannot be analyzed by a binding model.

In addition to MST, we now also provide ITC binding data with GTP γ S (Fig. S9) AND GMPPNP (Fig. S12), where the same tendency as derived from MST is observed, i.e. lower binding affinity with GMPPNP than GTP γ S (K_D : ~14 μ M versus 1.7 μ M).

Q9. Although computational methods have been employed to show that RACB-RIPb interactions are associated with membrane interactions, experimental evidence is required to support the hypothesis. Binding experiments of nucleotide-bound RACB with plant lipids in the presence and absence of RIPb will further the understanding of the structural basis to functional significance.

Thanks for this comment. The RACB-RIPb complex has been previously shown to be located at the plasma membrane (McCollum et al, *Plant Physiol*, 2020, therein Fig. S6). Furthermore, the lipid interaction of RACB in vitro has been characterized in Weiß et al, *Plant J*, 2025, which is now cited in our manuscript. In addition, the cellular fluorescence images in Fig. 6a show a strong co-localization of RACB and RIPb at the plasma membrane, further indicating that the complex localizes to the plasma membrane.

A: Thanks to all three reviewers for their final positive assessment of our work. We indeed underwent substantial efforts to make this manuscript a suitable candidate for publication by fully addressing all comments raised by all three reviewers.

Below, please find our response to the minor comment raised by reviewer #2.

Reviewer #2 (Remarks to the Author):

R2: The manuscript has been reviewed in considerable depth by three reviewers and I find that my comments and it appears on quick reading also those of the other reviewers have been adequately addressed.

A: Thanks for this positive summary.

One minor comment: when gel-filtration is mentioned as evidence for non-aggregation please estimate and state the protein concentration on the column by absorbance.

A: Thanks for this comment. We now have calculated the peak protein concentration on the SEC column and report these values now in the figure legend to Supplementary Fig. 2. The concentration range in SEC is also mentioned in the main text where Supplementary Fig. 2 is referenced in the NMR relaxation section.

For the analytical SEC runs to probe the interaction between RACB and RIPb-CC2 shown in Supplementary Fig. 12, we used protein concentrations of 20 μ M each, as already reported the legend to Supplementary Fig. 12. In this case, we did not calculate peak concentrations since both proteins are co-eluting in a complex thus making a quantification difficult, and also not necessary at this point.